# Bigger, Regularized, Optimistic: scaling for compute and sample-efficient continuous control

**Michal Nauman**[1,2]  **Mateusz Ostaszewski**[3]  **Krzysztof Jankowski**[2]  **Piotr Miłoś**[1,2,4]

**Marek Cygan**[2,5]

## Abstract

Sample efficiency in Reinforcement Learning (RL) has traditionally been driven by algorithmic enhancements. In this work, we demonstrate that scaling can also lead to substantial improvements. We conduct a thorough investigation into the interplay of scaling model capacity and domain-specific RL enhancements. These empirical findings inform the design choices underlying our proposed BRO (Bigger, Regularized, Optimistic) algorithm. The key insight behind BRO is that strong regularization allows for effective scaling of the critic networks, which, paired with optimistic exploration, leads to superior performance. BRO achieves state-of-the-art results, significantly outperforming the leading model-based and model-free algorithms across 40 complex tasks from the DeepMind Control, MetaWorld, and MyoSuite benchmarks. BRO is the first model-free algorithm to achieve near-optimal policies in the notoriously challenging Dog and Humanoid tasks.

## 1 Introduction

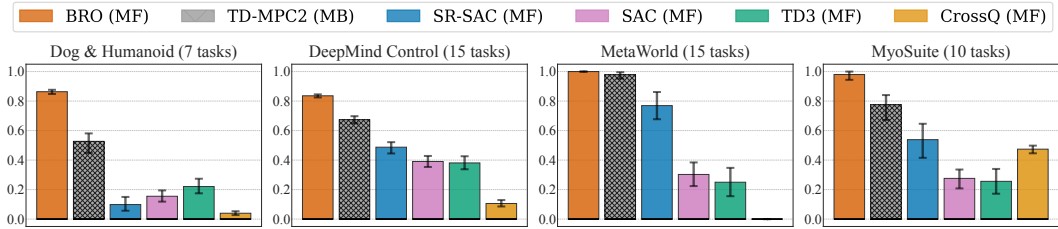

Figure 1: BRO sets new state-of-the-art outperforming model-free (MF) and model-based (MB) algorithms on 40 complex tasks covering 3 benchmark suites. Y-axes report interquartile mean calculated on 10 random seeds, with 1.0 representing the best possible performance in a given benchmark. We use $1M$ environment steps.

Deep learning has seen remarkable advancements in recent years, driven primarily by the development of large neural network models (Devlin et al., 2019; Tan & Le, 2019; Dosovitskiy et al., 2020). These advancements have significantly benefited fields like natural language processing and computer vision and have been percolating to RL as well (Padalkar et al., 2023; Zitkovich et al., 2023). Interestingly, some recent work has shown that the model scaling can be repurposed to achieve sample efficiency in discrete control (Schwarzer et al., 2023; Obando-Ceron et al., 2024), but these approaches cannot be directly translated to continuous action RL. As such, they rely on discrete action representation, whereas many physical control tasks have continuous, real-valued action spaces.

[1]Ideas NCBR; [2]University of Warsaw, [3]Warsaw University of Technology, [4]Polish Academy of Sciences, [5]Nomagic. Correspondence to: Michal Nauman <nauman.mic@gmail.com>

38th Conference on Neural Information Processing Systems (NeurIPS 2024).

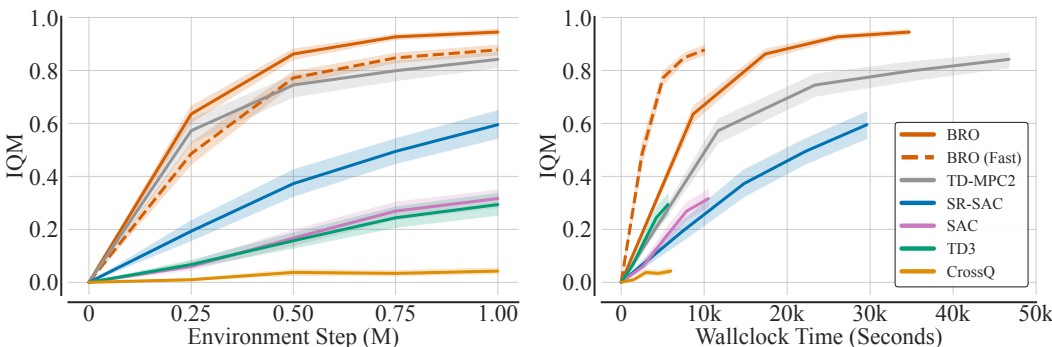

Figure 2: We report sample efficiency (left) and wallclock time (right) for BRO and BRO (Fast) (BRO with reduced replay ratio for increased compute efficiency), as well as baseline algorithms averaged over 40 tasks listed in Table 4. BRO achieves the best sample efficiency, whereas BRO (Fast) matches the sample efficiency of model-based TD-MPC2. In terms of wall clock efficiency, BRO runs approximately 25% faster than TD-MPC2. Remarkably, BRO (Fast) matches the wallclock efficiency of a standard SAC agent while achieving 400% better performance. The Y-axis reports the interquartile mean, with 1.0 representing the maximal possible performance.

Conventional practice in continuous deep RL has relied on small network architectures (Haarnoja et al., 2018; Hiraoka et al., 2021; Raffin et al., 2021; D'Oro et al., 2022), with the primary focus on algorithmic improvements. These enhancements aim to achieve better sample efficiency and address key challenges such as value overestimation (Fujimoto et al., 2018; Moskovitz et al., 2021; Cetin & Celiktutan, 2023), exploration (Chen et al., 2017; Ciosek et al., 2019; Nauman & Cygan, 2023a), and increasing the number of gradient steps (Nikishin et al., 2022; D'Oro et al., 2022). Additionally, evidence suggests that naive model capacity scaling can degrade performance (Andrychowicz et al., 2021; Bjorck et al., 2021). We challenge this status quo by posing a critical question: *Can significant performance improvements in continuous control be achieved by combining parameter and replay scaling with existing algorithmic improvements?*

In this work, we answer this question affirmatively, identifying components essential to successful scaling. Our findings are based on a thorough evaluation of a broad range of design choices, which include batch size (Obando Ceron et al., 2024), distributional Q-values techniques (Bellemare et al., 2017; Dabney et al., 2018), neural network regularizations (Bjorck et al., 2021; Nauman et al., 2024), and optimistic exploration (Moskovitz et al., 2021; Nauman & Cygan, 2023a). Moreover, we carefully investigate the benefits and computational costs stemming from scaling along two axes: the number of parameters and the number of gradient steps. Importantly, we find that the former can lead to more significant performance gains while being more computationally efficient in parallelized setups.

Our work culminates in developing the BRO (Bigger, Regularized, Optimistic) algorithm, a novel sample-efficient model-free approach. BRO significantly outperforms existing model-free and model-based approaches on 40 demanding tasks from the DeepMind Control, MetaWorld, and MyoSuite benchmarks, as illustrated in Figures 1 and 2. Notably, BRO is the first model-free algorithm to achieve near-optimal performance in challenging Dog and Humanoid tasks while being 2.5 times more sample-efficient than the leading model-based algorithm, TD-MPC2. The key BRO innovation is pairing strong regularization with critic model scaling, which, coupled with optimistic exploration, leads to superior performance. We summarize our contributions:

- **Extensive empirical analysis** - we conduct an extensive empirical analysis focusing on critic model scaling in continuous deep RL. By training over 15,000 agents, we explore the interplay between critic capacity, replay ratio, and a comprehensive list of design choices.
- **BroNet architecture & BRO algorithm** - we introduce the BRO algorithm, a novel model-free approach that combines BroNet architecture for critic scaling with domain-specific RL enhancements. BRO achieves state-of-the-art performance on 40 challenging tasks across diverse domains.
- **Scaling & regularization** - we offer several insights, with the most important being: regularized critic scaling outperforms replay ratio scaling in terms of performance and computational efficiency; the inductive biases introduced by domain-specific RL improvements can be largely substituted by critic scaling, leading to simpler algorithms.

## 2 Bigger, Regularized, Optimistic (BRO) algorithm

This section presents our novel Big, Regularized, Optimistic (BRO) algorithm and its design principles. The model-free BRO is a conclusion of extensive experimentation presented in Section 3, and significantly outperforms existing state-of-the-art methods on continuous control tasks from proprioceptive states (Figure 1).

### 2.1 Experimental setup

We compare BRO against a variety of baseline algorithms. Firstly, we consider TD-MPC2 (Hansen et al., 2023), a model-based state-of-the-art that was shown to reliably solve the complex dog domains. Secondly, we consider SR-SAC (D'Oro et al., 2022), a sample-efficient SAC implementation that uses a large replay ratio of 32 and full-parameter resets. For completeness, we also consider CrossQ (Bhatt et al., 2023), a compute-efficient method that was shown to outperform ensemble approaches, as well as standard SAC (Haarnoja et al., 2018) and TD3 (Fujimoto et al., 2018). We run all algorithms with 10 random seeds, except for TD-MPC2, for which we use the results provided by the original manuscript (Hansen et al., 2023). We describe the process of hyperparameter selection for all considered algorithms in Appendix D, and share BRO pseudocode in Appendix (Pseudocode 1). We implement BRO based on the JaxRL (Kostrikov, 2021) and make the code available under the following link: https://github.com/naumix/BiggerRegularizedOptimistic

**Environments**   We consider a wide range of control tasks, encompassing a total of 40 diverse, complex continuous control tasks spanning three simulation domains: DeepMind Control (Tassa et al., 2018), MetaWorld (Yu et al., 2020), and MyoSuite (Caggiano et al., 2022) (a detailed list of environments can be found in Appendix C). These tasks include high-dimensional state and action spaces (with $|S|$ and $|A|$ reaching 223 and 39 dimensions), sparse rewards, complex locomotion tasks, and physiologically accurate musculoskeletal motor control. We run the algorithms for $1M$ environment steps and report the final performance unless explicitly stated otherwise. We calculate the interquartile means and confidence intervals using the RLiable package (Agarwal et al., 2021).

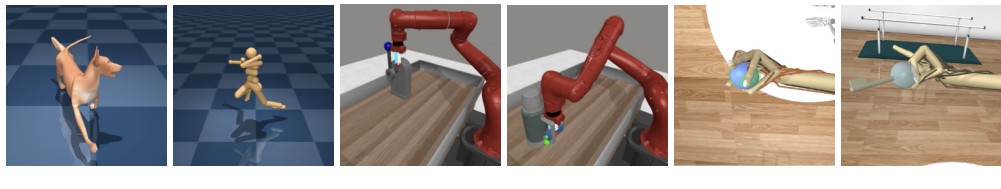

(a) DeepMind Control (DMC)          (b) MetaWorld (MW)          (c) MyoSuite (MS)

Figure 3: We consider a total of 40 tasks from DeepMind Control (DMC), MetaWorld (MW), and MyoSuite (MS). In particular, we chose the tasks with the biggest optimality gap according to previous evaluations (Hansen et al., 2023). We list all considered tasks in Table 4.

### 2.2 BRO outline and design choices

The BRO algorithm is based on the well-established Soft Actor-Critic (SAC) (Haarnoja et al., 2018) (see also Appendix A) and is composed of the following key components:

- **Bigger** – BRO uses a scaled critic network with the default of $\approx 5M$ parameters, which is approximately 7 times larger than the average size of SAC models (Haarnoja et al., 2018); as well as scaled training density with a default replay ratio[1] of $RR = 10$, and $RR = 2$ for the BRO (Fast) version.

- **Regularized** – the BroNet architecture, intrinsic to the BRO approach, incorporates strategies for regularization and stability enhancement, including the utilization of Layer Normalization (Ba et al., 2016) after each dense layer, alongside weight decay (Loshchilov & Hutter, 2017) and full-parameter resets (Nikishin et al., 2022).

---

[1]The replay ratio refers to the number of gradient updates per one environment step.

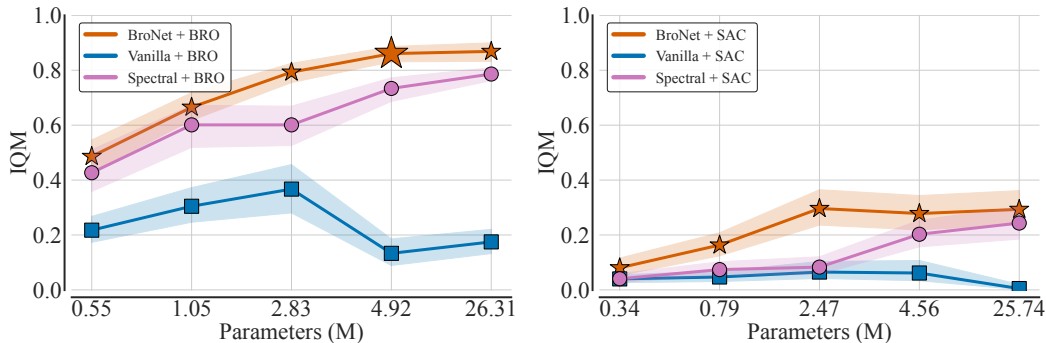

Figure 4: Scaling the critic parameter count for vanilla dense (Fujimoto et al., 2018), spectral normalization ResNet (Bjorck et al., 2021), and our BroNet for BRO (left), and SAC (right). We conclude that to achieve the best performance, we need both the right architecture (BroNet) and the correct algorithmic enhancements encapsulated in BRO. We report interquartile mean performance after $1M$ environment steps in tasks listed in Table 3, with error bars indicating 95% CI from 10 seeds. On the X-axis, we report the approximate parameter count of each configuration.

- **Optimistic** – BRO uses dual policy optimistic exploration (Nauman & Cygan, 2023a) and non-pessimistic (Nauman et al., 2024) quantile $Q$-value approximation (Dabney et al., 2018; Moskovitz et al., 2021) for balancing exploration and exploitation.

The full details of the algorithm, along with the pseudo-code, are provided in Appendix B. Figure 7 summarizes the impact of removing components of BRO. We observe the biggest impact of scaling the critic capacity (scale) and replay ratio ($RR$), as well as using non-pessimistic $Q$-value, i.e. removing Clipped Double $Q$-learning (CDQ).

**Scaling critic network and BroNet architecture** The key contribution of this paper is showing how to enable scaling the critic network. We recall that naively increasing the critic capacity does not necessarily lead to performance improvements and that successful scaling depends on a carefully chosen suite of regularization techniques (Bjorck et al., 2021). Figure 5 shows our BroNet architecture, which, up to our knowledge, did not exist previously in the literature. The architecture begins with a dense layer followed by Layer Norm (Ba et al., 2016) and ReLU activation. Subsequently, the network comprises ResNet blocks, each consisting of two dense layers regularized with Layer Norm. Consequently, the ResNet resembles the FFN sub-layer utilized in modern LLM architectures (Xiong et al., 2020), differing primarily in the placement of the Layer Norms. Crucially, we find that BroNet scales more effectively than other architectures (Figure 4 (left)). However, the right choice of architecture and scaling is not a silver bullet. Figure 4 (right) shows that when these are plugged into the standard SAC algorithm naively, the performance is weak. The important elements are additional regularization

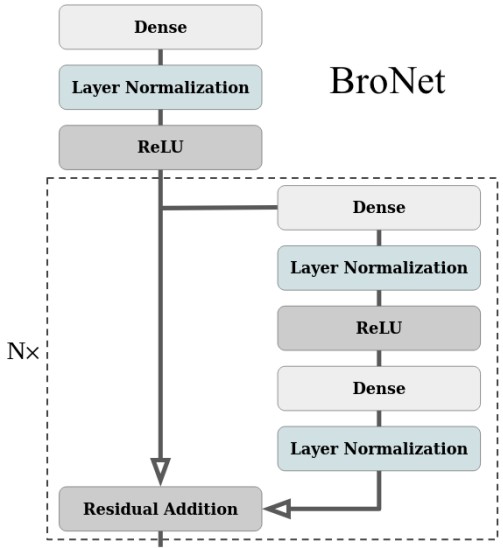

Figure 5: BroNet architecture employed for actor and critic. Each fully connected layer is augmented with Layer Norm, which is essential to unlocking scaling. We use $\approx 5M$ parameters and $N = 2$ in the default setting.

(weight decay and network resets) and optimistic exploration (see below). Interestingly, we did not find benefits from scaling the actor networks, further discussed in Section 3.

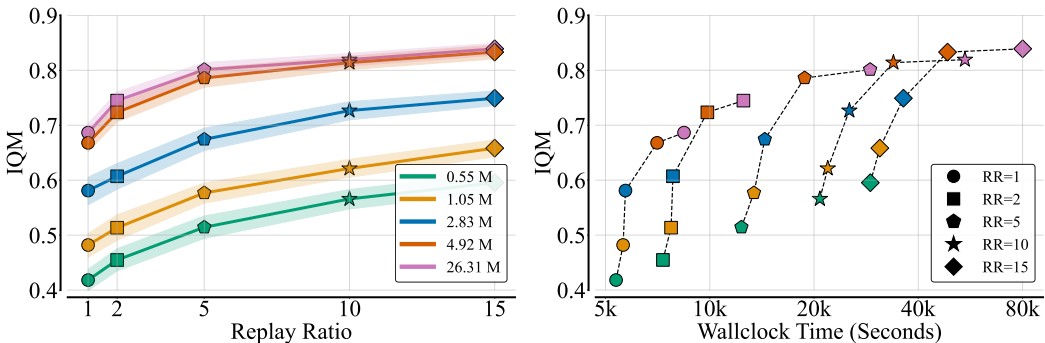

Figure 6: To account for sample efficiency, we report the performance averaged at $250k$, $500k$, $750k$, and $1M$ environment steps across different 5 replay ratios and 5 critic model sizes. All agents were evaluated in tasks listed in Table 3, and 10 random seeds per variant. The left figure shows performance scaling with increasing replay ratios (shapes) and model sizes (colors). The right figure examines the tradeoff between performance and computational cost when scaling replay ratios versus critic model sizes. Increasing model size leads to substantial performance improvements at lower compute costs compared to increasing the replay ratio. We present more scaling results in Appendix E, including a description of model sizes in Table 7.

**Scaling replay ratio and relation to model scaling**    Increasing replay ratio (D'Oro et al., 2022) is another axis of scaling. We investigate mutual interactions by measuring the performance across different model scales (from $0.55M$ to $26M$) and $RR$ settings (from $RR = 1$ to $RR = 15$). Figure 6 reveals that the model scaling has a much stronger impact plateauing at $\approx 5M$ parameters. Increasing the replay ratio also leads to noticeable benefits. For example, a $26M$ model with $RR = 1$ achieves significantly better performance than a small model with $RR = 15$, even though the $26M$ model requires three times less wallclock time. Importantly, model scaling and increasing replay ratio work well in tandem and are interchangeable to some degree. We additionally note that the replay ratio has a bigger impact on wallclock time than the model size. This stems from the fact that scaling replay ratio leads to inherently sequential calculations, whereas scaling model size leads to calculations that can be parallelized. For these reasons, BRO (Fast) with $RR = 2$ and $5M$ network offers an attractive trade-off, being already very sample efficient and fast at the same time.

**Optimistic exploration and Q-values**    BRO utilizes two mechanisms to increase optimism. We observe significant improvements stemming from these techniques in both BRO and BRO (Fast) agents (Figure 7). They are particularly pronounced in the early stages of the training and for smaller models (Figure 9a).

The initial mechanism involves deactivating Clipped Double Q-learning (CDQ) (Fujimoto et al., 2018), a commonly employed technique in reinforcement learning aimed at mitigating Q-value overestimation. For further clarification, refer to Appendix B.6, particularly Eq. 8 where we take the ensemble mean instead of minimum for Q-value calculation. This is surprising, perhaps, as it goes against conventional wisdom. However, some recent work has already suggested that regularization might effectively combat the overestimation (Nauman et al., 2024). We observe a much stronger effect. In Figures 7 & 9a, we compare the performance of BRO with BRO that uses CDQ. This analysis indicates that using risk-neutral Q-value approximation in the presence of network regularization unlocks significant performance improvements without value overestimation (Table 1).

The second mechanism is optimistic exploration. We implement the dual actor setup (Nauman & Cygan, 2023a), which employs separate policies for exploration and temporal difference updates. The exploration policy follows an optimistic upper-bound Q-value approximation, which has been shown to improve the sample efficiency of SAC-based agents (Ciosek et al., 2019; Moskovitz et al., 2021; Nauman & Cygan, 2023a). In particular, we optimize the optimistic actor towards a KL-regularized Q-value upper-bound (Nauman & Cygan, 2023a), with Q-value upper-bound calculated with respect to epistemic uncertainty calculated according to the methodology presented in Moskovitz et al. (2021). As shown in Figure 7, using dual actor optimistic exploration yields around $10\%$ performance improvement in the BRO model.

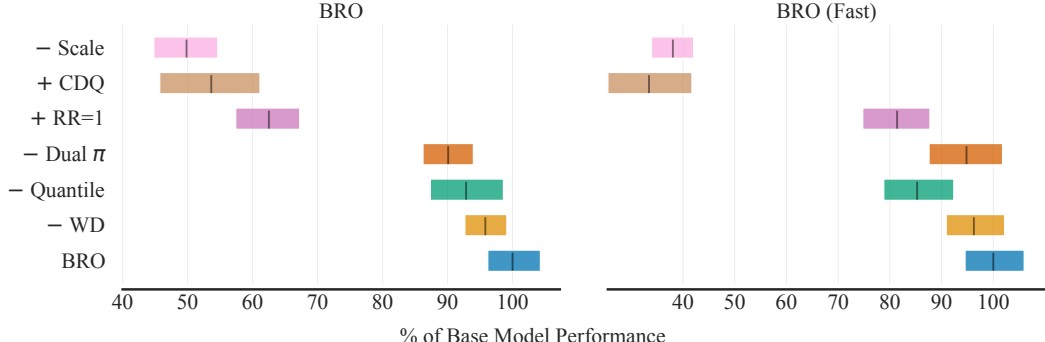

Figure 7: Impact of removing various BRO components on its performance. We report the percentage of the final performance for BRO (left) and BRO (Fast) (right). The y-axis shows the components that are ablated: **-Scale** denotes using a standard-sized network, **+CDQ** denotes using pessimistic Clipped Double Q-learning (which is removed by default in BRO), **+RR=1** uses the standard replay ratio, **-Dual** $\pi$ removes optimistic exploration, and **-Quantile** and **-WD** stand for removing quantile Q-values and weight decay, respectively. We report the interquartile mean and 95% CIs for tasks in Table 3, with 10 random seeds. The results indicate that the **Scale**, **CDQ**, and **RR=1** components are the most impactful for BRO. Since BRO (Fast) has RR=2 by default, reducing it to one does not significantly affect its performance.

**Others** We mention two other design choices. First, we use a smaller batch size of $128$ than the typical one of $256$. This is computationally beneficial while having a marginal impact on performance, which we show in Figure 15. Secondly, we use quantile Q-values (Bellemare et al., 2017; Dabney et al., 2018). We find that quantile critic representation improves performance (Figure 7), particularly for smaller networks. This improvement, however, diminishes for over-parameterized agents (Figure 9a). On top of the performance improvements, the distribution setup allows us to estimate epistemic uncertainties, which we leverage in the optimistic exploration according to the methodology presented in Moskovitz et al. (2021).

## 3 Analysis

This section summarizes the results of 15,000 experiments, detailed in Table 2, which led us to develop the BRO algorithms. These experiments also provided numerous insights that we believe will be of interest to the community. We adhered to the experimental setup described in Section 2.1. We also present additional experimental results in Appendix E.

**Scaling model-free critic allows superior performance** We recall that the most important finding is that skillful critic model scaling combined with simple algorithmic improvements can lead to extremely sample-efficient performance and the ability to solve the most challenging environments. We deepen these observations in experiments depicted in Figure 8. Namely, we let the other algorithms, including state-of-the-art model-based TD-MPC2, run for 3M steps on the most challenging tasks in the DMC suite (Dog Stand, Dog Walk, Dog Trot, Dog Run, Humanoid Stand, Humanoid Walk, and Humanoid Run). TD-MPC2 eventually achieves BRO performance levels, but it requires approximately $2.5$ more environment steps.

**Algorithmic improvements matter less as the scale increases** The impact of algorithmic improvements varies with the size of the critic model. As shown in Figure 9a, while techniques like smaller batch sizes, quantile Q-values, and optimistic exploration enhance performance for $1.05M$ and $4.92M$ models, they do not improve performance for the largest $26.3M$ models. We hypothesize this reflects a tradeoff between the inductive bias of domain-specific RL techniques and the overparameterization of large neural networks. Despite this, these techniques still offer performance gains with lower computing costs. Notably, full-parameter resets (Nikishin et al., 2022; D'Oro et al., 2022) are beneficial; the largest model without resets nearly matches the performance of the BRO with resets.

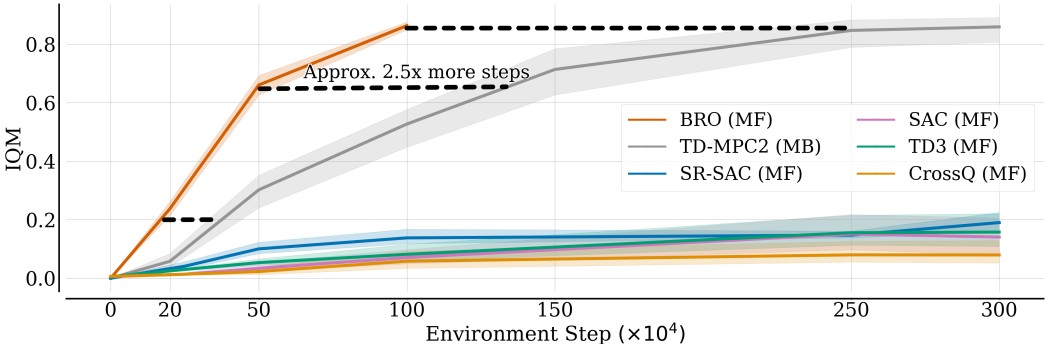

Figure 8: IQM return learning curves for four Dog and three Humanoid environments from the DMC benchmark, plotted against the number of environment steps. Notably, the model-based approach (TD-MPC2) requires approximately 2.5 times more steps to match BRO performance.

**Scaling actor is not effective** Previous works underscore the relative importance of critic and actor networks in off-policy algorithms like SAC (Fujimoto et al., 2018; D'Oro et al., 2022; Li et al., 2022). For instance, Nikishin et al. (2022) found that critic regularization is significantly more important than actor regularization. We confirm this result by showing that, for off-policy continuous control actor-critic algorithms, increasing critic capacity leads to much better results than increasing the actor model size, which in some cases might be even detrimental (Figure 9a). As such, practitioners can achieve performance improvements by prioritizing critic capacity over actor capacity while adhering to memory limitations.

**Target networks yield small but noticeable performance benefits** Using target networks doubles the memory costs (Schwarzer et al., 2020; Bhatt et al., 2023; Lee et al., 2024), which can be a significant burden for large models. In Figure 9b, we compare the performance of standard BRO and BRO (Fast) agents against their versions without target networks. Consistent with established understanding, we find that using target networks yields a small but significant performance improvement. However, we observe substantial variation in these effects among benchmarks and specific environments (Figure 9b & Figure 16). For example, the majority of performance improvements in DMC and MS environments are attributable to specific tasks.

**Architecture matters (especially in complex environments)** By breaking down the results from Figure 4 into individual environments, the BroNet architecture achieves better performance in all of them, but the differences are most pronounced in the Dog environments. Therefore, we deepened our analysis with extra metrics to understand these discrepancies better. Table 1 demonstrates that BroNet outperforms the other architectures regarding final performance.

Table 1: Comparison of BroNet, Spectral (Bjorck et al., 2021), and Vanilla MLP architectures in notriously hard Dog environments. All metrics except return are averaged over time steps. All architectures are combined with BRO.

|  | BroNet | Spectral | Vanilla |
|---|---|---|---|
| Final return | 763.5 | 73.5 | 167. |
| $\|\nabla\|_2$ | 35.5 | 88. | 9.61E+04 |
| Mean Q-values | 58.06 | 153.85 | 1.20E+05 |
| TD-error | 0.38 | 4.31E+04 | 6.03E+07 |

The Vanilla MLP exhibits instabilities across all measured metrics, including gradient norm, overestimation, and TD error. While using the Spectral architecture maintains moderate gradient norms and overestimation, it struggles significantly with minimizing the TD error.

In (Nauman et al., 2024), the authors indicate that the gradient norm and overestimation are strong indicators of poor performance in Dog environments. However, these results suggest that identifying a single cause for the challenges in training a reinforcement learning agent is difficult, highlighting the complexity of these systems and the multifaceted nature of their performance issues.

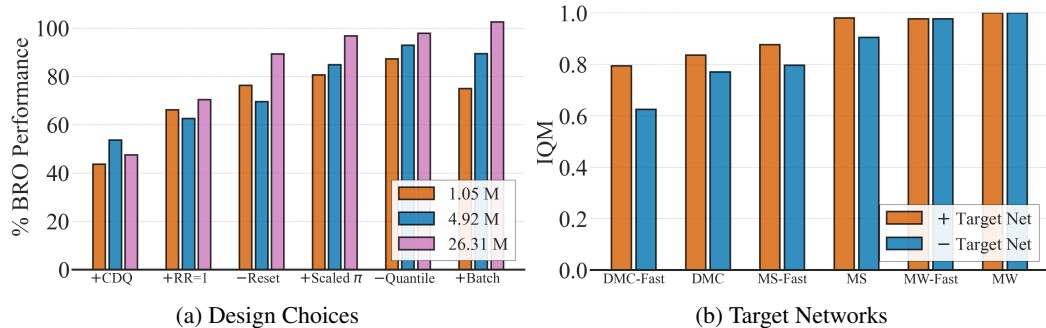

(a) Design Choices                  (b) Target Networks

Figure 9: (Left) We analyze the importance of BRO components dependent on the critic model size. Interestingly, most components become less important as the critic capacity grows. (Right) We report the performance of BRO variants with and without a target network. All algorithm variants are run with 10 random seeds.

**What did not work?** While researching BRO, we tested a variety of techniques that were found to improve the performance of different RL agents; however, they did not work in our evaluations. Firstly, we found that using $N$-step returns (Sutton & Barto, 2018; Schwarzer et al., 2023) does not improve the performance in the tested environments. We conjecture that the difference between $N$-step effectiveness in Atari and continuous control benchmarks stems from the sparser reward density in the former. Furthermore, we evaluated categorical RL (Bellemare et al., 2017) and HLGauss (Imani & White, 2018; Farebrother et al., 2024) Q-value representations, but found that these techniques are not directly transferable to a deterministic policy gradient setup and introduce training instabilities when applied naively, resulting in a significant amount of seeds not finishing their training. Finally, we tested a variety of scheduling mechanisms considered by Schwarzer et al. (2023) but found that the performance benefits are marginal and highly task-dependent while introducing much more complexity associated with hyperparameter tuning. A complete list of tested techniques is presented in Appendix B.8.

**Are current benchmarks enough?** As illustrated in Figure 10, even complex tasks like Dog Walk or Dog Trot can be reliably solved by combining existing algorithmic improvements with critic model scaling within 1 million environment steps. However, some tasks remain unsolved within this limit (e.g., Humanoid Run or Acrobot Swingup). Tailoring algorithms to single tasks risks overfitting to specific issues. Therefore, we advocate for standardized benchmarks that reflect the sample efficiency of modern algorithms. This standardization would facilitate consistent comparison of approaches, accelerate advancements by focusing on a common set of challenging tasks, and promote the development of more robust and generalizable RL algorithms. On that note, in Appendix F, we report BRO performance at earlier stages of the training.

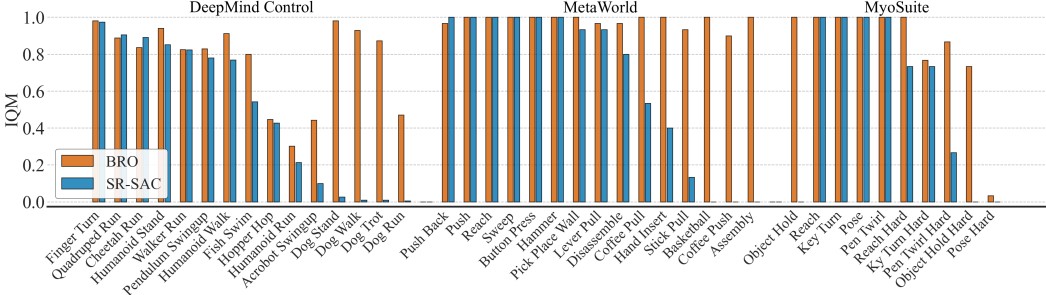

Figure 10: Our experiments cover 40 of the hardest tasks from DMC (locomotion), MW (manipulation), and MS (physiologically accurate musculoskeletal control) considered in previous work (Hansen et al., 2023). In those tasks, the state-of-the-art model-free SR-SAC (D'Oro et al., 2022) achieves more than 80% of maximal performance in 18 out of 40 tasks, whereas our proposed BRO in 33 out of 40 tasks. BRO makes significant progress in the most complex tasks of the benchmarks.

**BroNet architecture is useful beyond continuous control**  We design additional experiments to test the effectiveness of the naive application of BroNet to popular offline reinforcement learning problems in two offline RL benchmarks: AntMaze (6 tasks); and Adroit (9 tasks) (Fu et al., 2020). We run Behavioral Cloning (BC) in pure offline (Sutton & Barto, 2018), Implicit Q-Learning (IQL) offline + fine-tuning (Kostrikov et al., 2022), as well as online reinforcement learning with offline data (Ball et al., 2023). We run all these algorithms with the default MLP network, as well as BroNet backbone. As shown in Figure 11, we find that the naive application of BroNet leads to performance improvements across all tested algorithms.

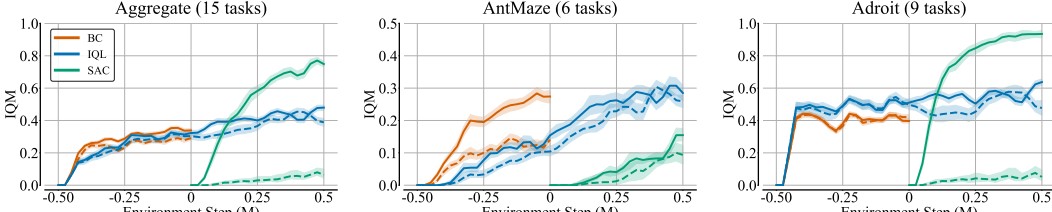

Figure 11: We test three scenarios: offline (comparing vanilla BC to BroNet-based BC), offline fine-tuning (comparing vanilla IQL to BroNet-based IQL), and online with offline data (comparing vanilla SAC to BroNet-based SAC). The solid line represents BRO-based and the dashed line represents vanilla variants. Negative values on the X-axis refer to offline training. 10 seeds per task.

# 4   Related Work

## 4.1   Sample efficiency through algorithmic improvements

A significant effort in RL has focused on algorithmic improvements. One recurring theme is controlling value overestimation (Fujimoto et al., 2018; Moskovitz et al., 2021; Cetin & Celiktutan, 2023). For instance, Fujimoto et al. (2018) proposed Clipped Double Q-learning (CDQ), which updates policy and value networks using a lower-bound Q-value approximation. However, since a pessimistic lower-bound can slow down learning, Moskovitz et al. (2021) introduced an approach that tunes pessimism online. Recently, Nauman et al. (2024) showed that layer normalization can improve performance without value overestimation, eliminating the need for pessimistic Q-learning.

A notable effort has also focused on optimistic exploration (Wang et al., 2020; Moskovitz et al., 2021). Various methods have been developed to increase sample efficiency via exploration that is greedy with respect to a Q-value upper bound. These include closed-form transformations of the pessimistic policy (Ciosek et al., 2019) or using a dual actor network dedicated to exploration (Nauman & Cygan, 2023a).

## 4.2   Sample efficiency through scaling

Recent studies demonstrated the benefits of model scaling when pre-training on large datasets (Driess et al., 2023; Schubert et al., 2023; Taiga et al., 2023) or in pure offline RL setups (Kumar et al., 2023). Additionally, model scaling has proven advantageous for model-based online RL (Hafner et al., 2023; Hansen et al., 2023; Wang et al., 2024). However, in these approaches, most of the model scale is dedicated to world models, leaving the value network small. Notably, Schwarzer et al. (2023) found that increasing the scale of the encoder network improves performance for DQN agents, but did not study increasing the capacity of the value network. Various studies indicate that naive scaling of the value model leads to performance deterioration (Bjorck et al., 2021; Obando-Ceron et al., 2024; Farebrother et al., 2024). For example, Bjorck et al. (2021) demonstrated that spectral normalization enables stable training with relatively large ResNets with performance improvements.

In addition to model size scaling, the community has investigated the effectiveness of replay ratio scaling (i.e., increasing the number of gradient steps for every environment step) (Hiraoka et al., 2021; Nikishin et al., 2022; Li et al., 2022). Recent works have shown that a high replay ratio can improve performance across various algorithms in both continuous and discrete MDPs, provided the neural networks are regularized (Li et al., 2022; D'Oro et al., 2022). In this context, layer normalization

and full-parameter resets have been particularly effective (Schwarzer et al., 2023; Lyle et al., 2024; Nauman et al., 2024).

## 5  Limitations and Future Work

BRO's larger model size compared to traditional baselines like SAC or TD3 results in higher memory requirements, potentially posing challenges for real-time inference in high-frequency control tasks. Future research could explore techniques such as quantization or distillation to improve inference speed. While BRO is designed for continuous control problems, its effectiveness in discrete settings remains unexplored. Further investigation is needed to assess the applicability and performance of BRO's components in discrete action MDPs. Additionally, our experimentation primarily focuses on continuous control tasks using proprioceptive state representations. Future research is needed to investigate the tradeoff between scaling the critic and the state encoder in image-based RL.

## 6  Conclusions

Our study underscores the efficacy of scaling a regularized critic model in conjunction with existing algorithmic enhancements, resulting in sample-efficient methods for continuous-action RL. The proposed BRO algorithm achieves markedly superior performance within 1 million environment steps compared to the state-of-the-art model-based TD-MPC2 and other model-free baselines. Notably, it achieves over 90% success rates in MetaWorld and MyoSuite benchmarks, as well as over 85% of maximal returns in the DeepMind Control Suite, and near-optimal policies in the challenging Dog and Humanoid locomotion tasks. While some tasks remain unsolved within 1 million environment steps, our findings underscore the need for new standardized benchmarks focusing on sample efficiency to drive consistent progress in the field. The BRO algorithm establishes a new standard for sample efficiency, providing a solid foundation for future research to build upon and develop even more robust RL algorithms.

**Acknowledgements**

This research was partially supported by the National Science Centre, Poland, under grant nos. 2020/39/B/ST6/01511 and 2023/51/D/ST6/01609, as well as by the Warsaw University of Technology and the University of Warsaw through the Excellence Initiative: Research University (IDUB) program. We also gratefully acknowledge the Polish high-performance computing infrastructure, PLGrid (HPC Center: ACK Cyfronet AGH), for providing computational resources and support under grant no. PLG/2024/017159. Marek Cygan was partially supported by an NCBiR grant POIR.01.01.01-00-0433/20. Piotr Miłoś research was supported by the National Science Center (Poland) grant number 2019/35/O/ST6/03464.

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

**Broader Impact**

The work presented in this study, while academic and based on simulated benchmarks, advances the development of more capable autonomous agents. Although our contributions do not directly cause any negative societal impacts, we encourage the community to remain mindful of such potential consequences when extending our research.

# A   Background

We consider an infinite-horizon Markov Decision Process (MDP) (Puterman, 2014) which is described with a tuple $(S, A, r, p, \gamma)$, where states $S$ and actions $A$ are continuous, $r(s', s, a)$ is the transition reward, $p(s'|s, a)$ is the transition kernel and $\gamma \in (0, 1]$ is the discount factor.

The policy $\pi(a|s)$ is a state-conditioned action distribution with its entropy denoted as $\mathcal{H}(\pi(s))$. Soft Value (Haarnoja et al., 2018) is the sum of expected discounted return and state entropies from following the policy at a given state

$$V^\pi(s) = \mathrm{E}_{a \sim \pi, s' \sim p} \left[ r(s', s, a) + \alpha \mathcal{H}(\pi(s)) + \gamma V^\pi(s') \right], \tag{1}$$

with $\alpha$ denoting the entropy temperature parameter. Q-value is the expected discounted return from performing an action and following the policy thereafter.

$$Q^\pi(s, a) = \mathrm{E}_{s' \sim p} \left[ r(s', s, a) + \gamma V^\pi(s') \right] \tag{2}$$

A policy is said to be optimal if it maximizes the expected value of the possible starting states $s_0$, such that $\dot\pi = \arg\max_{\pi \in \Pi} \mathrm{E}_{s_0 \sim p} V^\pi(s_0)$, with $\dot\pi$ denoting the optimal policy and $\Pi$ denoting the considered set of policies (e.g., Gaussian). Soft values and soft Q-values are related through the following equation:

$$V^\pi(s) = \mathrm{E}_{a \sim \pi} \left[ Q^\pi(s, a) - \log \pi(a|s) \right] \tag{3}$$

This relation is often approximated via a single action sampled according to the policy $a \sim \pi(s)$. In off-policy actor-critic, there is continuous gradient-based learning of both Q-values (critic) and the policy (actor). The critic parameters $\theta$ are updated by minimizing SARSA temporal-difference on transitions $T = (s, a, r, s')$, with $T$ being sampled from a replay buffer of transitions (Fujimoto et al., 2018; Haarnoja et al., 2018) according to:

$$\theta = \arg\min_\theta \mathrm{E}_{T \sim \mathcal{D}} \left( Q_\theta(s, a) - r(s', s, a) - \gamma V_{\bar\theta}(s) \right), \tag{4}$$

$$V_{\bar\theta}(s) = Q_{\bar\theta}(s', a') - \alpha \log \pi_\phi(a'|s')), \tag{5}$$

with $a' \sim \pi_\phi$. In this setup, $Q_\theta$ is the critic network, $Q_{\bar\theta}$ is the value target network, and $\mathcal{D}$ is the replay buffer (Mnih et al., 2015). $Q_\theta$ is trained to approximate the Q-value under the policy from which the bootstrap is sampled (Van Seijen et al., 2009; Sutton & Barto, 2018). The policy parameters $\phi$ are updated to seek locally optimal values approximated by the critic (Ciosek & Whiteson, 2020) according to:

$$\phi = \arg\max_\phi \mathrm{E}_{s \sim \mathcal{D}} \left( Q_\theta(s, a) - \alpha \log \pi_\phi(a|s) \right), \quad \text{with} \quad a \sim \pi_\phi. \tag{6}$$

# B   BRO additional details

## B.1   Base agent

BRO uses the well-established Soft Actor-Critic (SAC) (Haarnoja et al., 2018) as its base. SAC is a stochastic policy, maximum entropy algorithm (see Eq. 1) that employs online entropy temperature adjustment and an ensemble of two critic networks. SAC models the policy via a Tanh-transformed Gaussian distribution whose parameters are modeled by the actor network.

## B.2   Architecture details

In the proposed architecture, we adopt the transformer feedforward blueprint from Vaswani et al. (2017) with a novel layer normalization configuration, as shown in Figure 5. Dropout is omitted. All Dense layers in the BRO have a default width of $512$ units, with a linear layer at both the input and output stages. To increase the model's depth, we add new residual blocks exclusively. While a similar transformer backbone has been used in previous work (Bjorck et al., 2021), our design choices, detailed in Section E, led us to use layer normalization instead of spectral normalization.

### B.3 Scaling

In Figure 4, we examine the scaling capabilities of SAC and BRO agents using a vanilla dense network (Fujimoto et al., 2018; Haarnoja et al., 2018; Moskovitz et al., 2021), a spectral normalization ResNet (Bjorck et al., 2021; Cetin & Celiktutan, 2023), and a layer normalization ResNet inspired by previous work (Nauman et al., 2024). As shown in Figure 4, increasing the model capacity of a vanilla-dense agent can lead to performance degradation beyond a certain model size. However, for the regularized architectures, we observe behavior similar to the empirically observed scaling properties of supervised models, where increasing model size leads to diminishing returns in performance improvements. Furthermore, we find that the layer normalization ResNet achieves better scaling properties than the spectral normalization architecture. Interestingly, the BRO agent consistently outperforms the baseline SAC across all architectures and network sizes, suggesting an interaction between parameter scaling and other algorithmic design choices. The highest performing SAC agent achieves around 25% of maximal performance, whereas our proposed BRO agent achieves more than 90%. Given that the BRO agent performs similarly at 4.92 million and 26.31 million parameters, we use the smaller model to reduce the computational burden.

### B.4 Scaling replay ratio

Replay Ratio (RR) scaling in small models leads to diminishing performance increases as RR values rise, eventually plateauing at higher RRs (Nikishin et al., 2022; D'Oro et al., 2022). Unfortunately, increasing RR also results in linear increases in computing costs, as each gradient step must be calculated sequentially. This naturally becomes a burden as the model sizes increase. In Figure 6, we investigate the performance of the BRO agent across different model scales (from 0.55 million to 26.31 million parameters) and RR settings (from RR=1 to RR=15), measuring both performance and wall-clock efficiency. We find that with the BRO regularized critic architecture, critic scaling leads to performance and sample efficiency gains that match those of scaled RR. Scaling both RR and model size produces the best-performing agents. Interestingly, scaling the model size can lead to significant performance improvements even if RR is low while being more computationally efficient due to parallelized workloads (Figure 6). For example, a 5 million parameter BRO model with RR=1 outperforms a 1 million parameter BRO agent with RR=15 despite being five times faster in terms of wall-clock time. This observation challenges the notion that a sample-efficient RL algorithm must use high replay settings.

### B.5 Batch Size

Inspired by recent findings that reducing the batch size can result in significant performance gains for discrete action RL (Obando Ceron et al., 2024), we reduce the number of transitions used for gradient calculation from 256 to 128. As shown in Figures 9a & 15, this batch size reduction leads to a marginal improvement in aggregate performance and decreased memory requirements of the algorithm. Interestingly, we find that batch size significantly affects performance, with no single value performing best across all tasks.

### B.6 Risk-Neutral Temporal Difference

Using a pessimistic lower-bound Q-value approximation for actor-critic updates, known as Clipped Double Q-learning (CDQ) (Fujimoto et al., 2018; Haarnoja et al., 2018), is a popular method to counteract Q-value overestimation, though it introduces bias. Formally, it modifies the value estimate in Eq. 5 to a lower-bound estimation

$$V_\theta^{lb}(s) \approx Q_\theta^{lb}(s, a) - \alpha \log \pi_\phi(a|s), \quad a \sim \pi_\phi(s), \tag{7}$$

where $Q^{lb}\theta(s, a)$ is a lower-bound Q-value estimation derived from a critic ensemble, often using two networks (Fujimoto et al., 2018; Haarnoja et al., 2018)

$$Q_\theta^{lb}(s, a) = \min(Q_\theta^1(s, a), Q_\theta^2(s, a)). \tag{8}$$

Recent studies have shown that techniques like layer normalization or full-parameter resets can be more effective at combating overestimation than pessimistic Q-value approximation (Nauman et al., 2024). Since our critic architecture leverages multiple regularization techniques, we disable CDQ and use the ensemble mean instead of the minimum to calculate the targets for actor and critic

updates. In Figures 7 & 9a, we compare the performance of the baseline BRO to a BRO agent that uses CDQ. Our findings indicate that using risk-neutral Q-value approximation in the presence of network regularization unlocks significant performance improvements without increasing value overestimation.

## B.7 Optimistic Exploration

Optimism is an algorithmic design principle that balances exploration and exploitation (Ciosek et al., 2019; Moskovitz et al., 2021). The dual actor setup (Nauman & Cygan, 2023a,b) employs separate policies for exploration and temporal difference updates, with the exploration policy pursuing an optimistic upper-bound Q-value approximation. This approach has been shown to improve the sample efficiency of SAC-based agents (Nauman & Cygan, 2023a). We implement the optimistic policy such that the Q-value upper bound is calculated based on the epistemic uncertainty estimated via the quantile critic ensemble (Moskovitz et al., 2021). Figure 7 shows that using a dual policy setup leads to performance improvements. We observe that these results are particularly pronounced in the early training stages and for smaller networks (Figure 9a).

---

**Algorithm 2** BRO training step with changes with respect to standard SAC colored red.

---

1: **Input:** $\pi_\phi^p$ - pessimistic actor; $\pi_\eta^o$ - optimistic actor; $Q_{\theta,i}^k$ - $k$th quantile of $i$th critic; $Q_{\bar\theta,i}^k$ - $k$th quantile of $i$th target critic; $\alpha$ - temperature; $\beta^o$ - optimism; $\tau$ - KL weight;

2: **Hyperparameters:** $\mathcal{KL}^*$ - target KL; $K$ - number of quantiles

---

3: *Sample action from the optimistic actor*
   $s', r = \text{ENV.STEP}(a) \quad a \sim \pi_\eta^o$
4: *Add transition to the replay buffer*
   $\text{BUFFER.ADD}(s, a, r, s')$
5: **for** $i = 1$ **to** ReplayRatio **do**
6:     *Sample batch of transitions*
   $s, a, r, s' \sim \text{BUFFER.SAMPLE}$
7:     *Calculate critic target value without CDQ*
   $Q_{\bar\theta}^\mu(s', a') = \frac{1}{2}(Q_{\bar\theta,1}^k(s', a') + Q_{\bar\theta,2}^k(s', a'))$    with    $a' \sim \pi_\phi^p(s')$
8:     *Update critic using pessimistic actor*
   $\theta \leftarrow \theta - \nabla_\theta Huber\big(Q_\theta(s, a), (r + \gamma Q_{\bar\theta}^\mu(s', a') - \alpha \log \pi_\phi^p(a'|s'))\big)$
9:     *Calculate pessimistic actor value without CDQ*
   $Q_\theta^\mu(s, a, \pi_\phi^p) = \frac{1}{2K} \sum_{i=1}^K (Q_{\theta,1}^k(s, a) + Q_{\theta,2}^k(s, a))$    with    $a \sim \pi_\phi^p(s)$
10:    *Update pessimistic actor*
   $\phi \leftarrow \phi + \nabla_\phi \big(Q_\theta^\mu(s, a, \pi_\phi^p) - \alpha \log \pi_\phi^p(a|s)\big)$
11:    *Calculate optimistic actor value*
   $Q_\theta^\mu(s, a, \pi_\eta^o) = \frac{1}{2K} \sum_{i=1}^K (Q_{\theta,1}^k(s, a) + Q_{\theta,2}^k(s, a) + \beta^o |Q_{\theta,1}^k(s, a) - Q_{\theta,2}^k(s, a)|), \; a \sim \pi_\eta^o(s)$
12:    *Update optimistic actor*
   $\eta \leftarrow \eta + \nabla_\eta \big(Q_\theta^\mu(s, a, \pi_\eta^o) + \beta^o Q_\theta^\sigma(s, a) - \tau KL\big(\pi_\phi^p(s)|\pi_\eta^o(s)\big)\big), \quad a \sim \pi_\eta^o(s)$
13:    *Update entropy temperature*
   $\alpha \leftarrow \alpha - \nabla_\alpha \alpha \big(\mathcal{H}^* - \mathcal{H}(s)\big)$
14:    *Update optimism*
   $\beta^o \leftarrow \beta^o - \nabla_{\beta^o}(\beta^o - \beta^p)(\frac{1}{|A|} KL(\pi_\phi^p|\pi_\eta^o) - \mathcal{KL}^*)$
15:    *Update KL weight*
   $\tau \leftarrow \tau + \nabla_\tau \tau(\frac{1}{|A|} KL(\pi_\phi^p|\pi_\eta^o) - \mathcal{KL}^*)$
16:    *Update target network*
   $\bar\theta \leftarrow \text{POLYAK}(\theta, \bar\theta)$
17: **end for**

---

## B.8 Approaches Examined During the Development of BRO

Examined approaches are listed in Table 2. Methods incorporated into BRO include regularization techniques (LayerNorm, Weight Decay, removing CDQL), optimistic exploration, quantile distributional RL, resets and increased replay ratio.

Table 2: Approaches examined during BRO development. Methods incorporated into BRO are highlighted in bold.

| Methods Group | Specific Method | Source |
|---|---|---|
| Exploration | **DAC** | (Nauman & Cygan, 2023a) |
| Value Regularization | CDQL **(removed)** | (Fujimoto et al., 2018) |
| | N-Step Returns | (Sutton & Barto, 2018) |
| Network Regularization | **LayerNorm** | (Ba et al., 2016) |
| | **Weight Decay** | (Loshchilov & Hutter, 2017) |
| | Spectral Norm | (Miyato et al., 2018) |
| Scheduling | N-Step Schedule | (Kearns & Singh, 2000) |
| | Discount Schedule | (François-Lavet et al., 2015) |
| | Pessimism Schedule | |
| | Entropy Schedule | |
| | Learning Rate Schedule | (Andrychowicz et al., 2021) |
| Distributional RL | HL Gauss | (Imani & White, 2018) |
| | Categorical | (Bellemare et al., 2017) |
| | **Quantile** | (Dabney et al., 2018) |
| Plasticity Regularization | **Resets** | (Nikishin et al., 2022; D'Oro et al., 2022) |
| Learning | **Replay Ratio** | (Nikishin et al., 2022; D'Oro et al., 2022) |

# C  Tested Environments

We tested BRO on a variety of 40 tasks from DeepMind Control Suite (Tassa et al., 2018), MyoSuite (Caggiano et al., 2022) and MetaWorld (Yu et al., 2020). Selected tasks cover various challenges, from simple to hard, in locomotion and manipulation. Table 4 presents the environments with specified dimensions of states and actions. BRO is a versatile agent that can successfully perform tasks of different difficulty and various action and state spaces. Our selection of 40 tasks focuses on the most challenging tasks from the DeepMind Control Suite (DMC), MetaWorld (MW), and MyoSuite (MS) benchmarks, as identified in previous studies (Hansen et al., 2023). We chose these hard tasks because many easy tasks from these benchmarks can be solved by modern algorithms within 100k environment steps (Hansen et al., 2023; Wang et al., 2024). In the MetaWorld environment, we follow the TD-MPC2 evaluation protocol (Hansen et al., 2023). As such, the environment issues a truncate signal after 200 environment steps, after which we assess if the agent achieved goal success within the 200th step. We do not implement any changes to how goals are defined in the original MetaWorld and we use V2 environments.

Table 3: List of tasks from DeepMind Control and MetaWorld on which the agents were ablated. The table also contains the dimensions of action and observation space.

| Task | Observation dimension | Action dimension |
|---|---|---|
| DEEPMIND CONTROL | | |
| Acrobot-Swingup | 6 | 1 |
| Dog-Trot | 223 | 38 |
| Hopper-Hop | 15 | 4 |
| Humanoid-Run | 67 | 24 |
| Humanoid-Walk | 67 | 24 |
| METAWORLD | | |
| Assembly | 39 | 4 |
| Coffee-Push | 39 | 4 |
| Hand-Insert | 39 | 4 |
| Push | 39 | 4 |
| Stick-Pull | 39 | 4 |

Table 4: List of tasks from DeepMind Control, MetaWorld, and MyoSuite on which the agents were tested. The table also contains the dimensions of action and observation space.

| Task | Observation dimension | Action dimension |
|------|:---:|:---:|
| DEEPMIND CONTROL | | |
| Acrobot-Swingup | 6 | 1 |
| Cheetah-Run | 17 | 6 |
| Dog-Run | 223 | 38 |
| Dog-Trot | 223 | 38 |
| Dog-Stand | 223 | 38 |
| Dog-Walk | 223 | 38 |
| Finger-Turn-Hard | 12 | 2 |
| Fish-Swim | 24 | 5 |
| Hopper-Hop | 15 | 4 |
| Humanoid-Run | 67 | 24 |
| Humanoid-Stand | 67 | 24 |
| Humanoid-Walk | 67 | 24 |
| Pendulum-Swingup | 3 | 1 |
| Quadruped-Run | 78 | 12 |
| Walker-Run | 24 | 6 |
| METAWORLD | | |
| Basketball | 39 | 4 |
| Assembly | 39 | 4 |
| Button-Press | 39 | 4 |
| Coffee-Pull | 39 | 4 |
| Coffee-Push | 39 | 4 |
| Disassemble | 39 | 4 |
| Hammer | 39 | 4 |
| Hand-Insert | 39 | 4 |
| Push | 39 | 4 |
| Reach | 39 | 4 |
| Stick-Pull | 39 | 4 |
| Sweep | 39 | 4 |
| Lever-Pull | 39 | 4 |
| Pick-Place | 39 | 4 |
| Push-Back | 39 | 4 |
| MYOSUITE | | |
| Key-Turn | 93 | 39 |
| Key-Turn-Hard | 93 | 39 |
| Obj-Hold | 91 | 39 |
| Obj-Hold-Hard | 91 | 39 |
| Pen-Twirl | 83 | 39 |
| Pen-Twirl-Hard | 83 | 39 |
| Pose | 108 | 39 |
| Pose-Hard | 108 | 39 |
| Reach | 115 | 39 |
| Reach-Hard | 115 | 39 |

# D  Hyperparameters

Hyperparameters of BRO and other baselines are listed in Table 5. BRO (Fast) shares the same parameters as BRO except replay ratio 2 which significantly speeds the algorithm without sacrificing performance that much. BRO features the BRONet architecture and resets of all parameters done every $250k$ steps until $1M$ steps with additional resets at steps $15k$ and $50k$. The selection of

hyperparameters for BRO was based on the values reported in the main building blocks of BRO and extensive experimentation coupled with ablations studies.

Table 5: Hyperparameter values for actor-critic agents used in the experiments.

| Parameter | BRO | SAC | TD3 | SR-SAC | CrossQ |
|---|---|---|---|---|---|
| Batch size ($B$) | 128 | 256 | | | |
| Replay ratio | 10 | 2 | | 32 | 1 |
| Critic hidden depth | RESIDUAL | 2 | | 2 | 2 |
| Critic hidden size | 512 | 256 | | | 2048 |
| Actor depth | RESIDUAL | 2 | | 2 | 2 |
| Actor size | | 256 | | | |
| Num quantiles | 100 | N/A | | | |
| KL target | 0.05 | N/A | | | |
| Initial optimism | 1.0 | N/A | | | |
| Std multiplier | 0.75 | N/A | | | |
| Actor learning rate | | 3e-4 | | | 1e-3 |
| Critic learning rate | | 3e-4 | | | 1e-3 |
| Temperature learning rate | 3e-4 | N/A | | 3e-4 | |
| Optimizer | ADAMW | ADAM | | | |
| Discount ($\gamma$) | | 0.99 | | | |
| Initial temperature ($\alpha_0$) | 1.0 | N/A | | 1.0 | |
| Exploratory steps | 2,500 | 10,000 | 25,000 | 10,000 | 5,000 |
| Target entropy ($\mathcal{H}^*$) | $|\mathcal{A}|/2$ | N/A | | $|\mathcal{A}|/2$ | $|\mathcal{A}|$ |
| Polyak weight ($\tau$) | | 0.005 | | | N/A |

We compare BRO against official and widely used implementations of CrossQ, SAC, SR-SAC, TD3 and TD-MPC2 with open source repositories listed in Table 6. As the official results do not cover all 40 benchmarking tasks, we ran the baselines independently (except TD-MPC2, where all official results were available). SAC and TD3 are commonly used baselines; therefore, their hyperparameters vary across different implementations. To account for this fact, we ran 2 versions of these baselines: tuned and original. If not specified otherwise, we report the results of the tuned versions with hyperparameters in Table 5. The original versions of SAC and TD3 both feature a replay ratio of 1 and in the case of SAC, target entropy ($\mathcal{H}^*$) equal to the action space dimension $|\mathcal{A}|$. The performance of both variants of the implementations can be observed in Figure 24.

Table 6: Links to the repositories of the used baselines. All are distributed under MIT license.

| Baseline | Source code link |
|---|---|
| CrossQ (Bhatt et al., 2023) | github.com/adityab/CrossQ |
| SAC (Haarnoja et al., 2018) | github.com/denisyarats/pytorch_sac |
| SAC (tuned version) | github.com/ikostrikov/jaxrl |
| SR-SAC (D'Oro et al., 2022) | github.com/proceduralia/high_replay_ratio_continuous_control |
| TD3 (Fujimoto et al., 2018) | github.com/sfujim/TD3 |
| TD3 (tuned version) | github.com/ikostrikov/jaxrl |
| TD-MPC2 (Hansen et al., 2023) | github.com/nicklashansen/tdmpc2 |

As other baselines were developed and tested on only a subset of our 40 selected tasks, we observed that achieving similar performance on new tasks was challenging. This can be especially observed in the case of CrossQ, which is a state-of-the-art algorithm on selected tasks from OpenAI Gym

(Brockman et al., 2016), but as it was tested only on a fraction of DeepMind Control Suite tasks, its performance does not transfer to our selection of tasks. Originally, CrossQ authors tested their agent on DeepMind Control Suite using Shimmy (Tai et al.) contrary to other agents that use the original codebase (Tassa et al., 2018). We run CrossQ using the DMC wrappers (D'Oro et al., 2022). Comparison between 2 variants of SAC and TD3 (Original and Tuned) is presented in Figure 24. Tuned versions feature a higher value of replay ratio (2 instead of 1) than the original and lower target entropy in the case of SAC ($|\mathcal{A}|/2$ instead of $|\mathcal{A}|$).

Table 7: Description of the considered model sizes.

| Size | Number of block | Hidden Size |
|---|---|---|
| $0.55M$ | 1 BroNet block | hidden size of 128 |
| $1.05M$ | 1 BroNet block | hidden size of 256 |
| $2.83M$ | 1 BroNet block | hidden size of 512 |
| $4.92M$ | 2 BroNet blocks | hidden size of 512 |
| $26.31M$ | 3 BroNet blocks | hidden size of 1024 |

# E    Additional Experiments

## E.1    Scaling and Time Experiments

The execution time was measured for all agents for each of the 40 tasks averaged over 2 random seeds. We ran each agent for $105k$ steps with initial $5k$ exploration steps, $100k$ training steps, and 1 evaluation. We benchmark all 25 variants of BRO with 5 different model sizes and 5 values of replay ratio. Figure 12 different algorithms performance compared to execution time. Experiments were conducted on an NVIDIA A100 GPU with 10GB of RAM and 8 CPU cores of AMD EPYC 7742 processor. All tasks were run separately so the agents could use all resources independently.

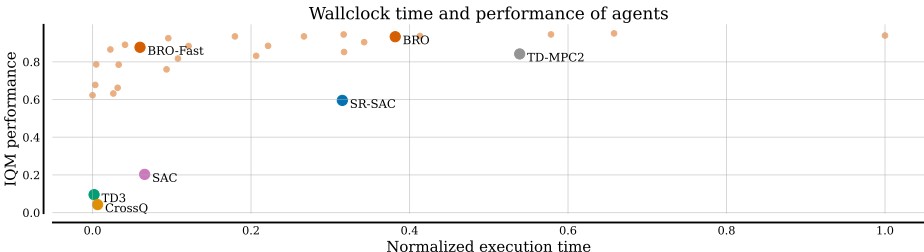

Figure 12: Scatterplot of the performance of agents plotted against normalized execution time.

Increasing the model size and replay ratio both improve the performance. However, the former is more efficient in terms of execution time due to GPU parallelism. For example, the largest BRO variant ($26.31M$ parameters) with replay ratio 5 has similar execution time as the smallest one ($0.55M$ parameters) with replay ratio 15, but the performance is much greater (Figures 13).

## E.2    Additional BroNet Analysis

We examine various architectural blueprints on 5 DMC and 5 MetaWorld environments (see Table 3), each with over 10 seeds per task. Our starting point was the transformer-based design by Bjorck et al. (2021), termed `Spectral`. This architecture incorporates recent transformer advancements, moving Layer Norm to the beginning of the residual block to prevent vanishing gradients in deep networks. While `Spectral` performs better than the vanilla MLP, its performance on the DMC benchmark, particularly the Dog environment, is weaker. This aligns with findings from Nauman et al. (2024); Hussing et al. (2024), indicating that Layer Norm is crucial for stability in such complex tasks. To analyze the importance of layer normalization in the BroNet architecture, we replaced spectral norms with Layer Norms in the residual blocks, resulting in the `BRO wo first LN` architecture (Figure 5). This modification improves performance but still lags behind the full BRO architecture. Furthermore,

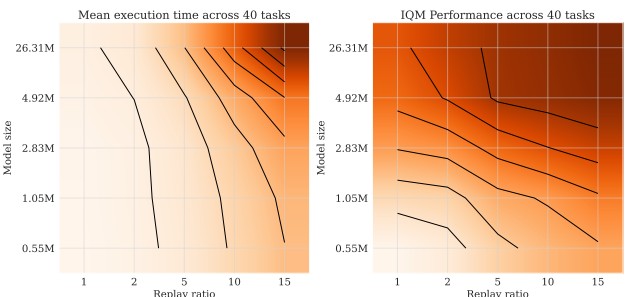

Figure 13: Heatmaps of execution time and IQM performance across 40 tasks of 25 BRO variants with various model sizes and replay ratio values. Black lines connect the same interpolated values.

we examine a simple MLP architecture with Layer Norm before each activation function. Since BRO consists of two residual blocks, we compare it with a 5-layer model, (Dense + LN) x 5. Figure 14 shows that Layer Norm after each Dense layer is effective, and in aggregated IQM, this model is comparable to BRO. However, skip connections in BRO are beneficial for managing complex environments like Dog. In conclusion, BroNet architecture uses Layer Norm and residual blocks for superior robustness and performance in challenging tasks.

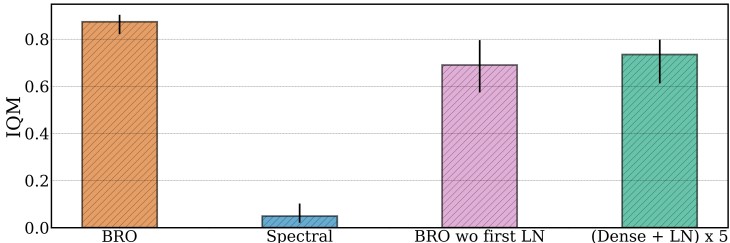

Figure 14: Comparison of five architecture designs across different environments: The top plot shows results on 5 DMC and 5 MetaWorld environments, the middle plot focuses on the 5 DMC environments, and the bottom plot highlights the Dog Trot environment. BRO and Spectral architectures each consist of 2 residual blocks. (Dense + LN) x 5 represents standard MLP networks with 5 hidden layers, each incorporating Layer Norm before activation. Lastly, BRO wo first LN refers to the BRO architecture without Layer Norm in the first Dense block, before the residual connection.

### E.3 Additional analysis

**Batch sizes** We ablate of the minibatch size impact on BRO and BRO (Fast) performance across different benchmarks is depicted in Figure 15. The figure shows that using half or even a quarter of the original minibatch size (256) does not significantly hurt BRO's performance.

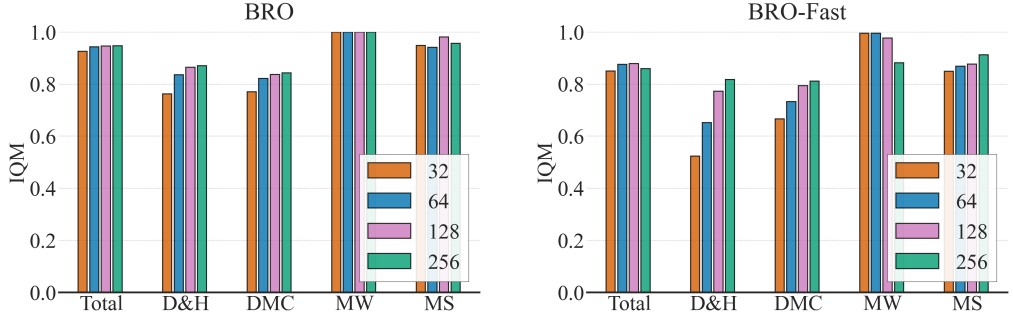

Figure 15: Performance of BRO and BRO (Fast) with different minibatch sizes for: D&H (Dogs and Humanoid), DMC (DeepMind Control), MW (MetaWorld), and MS (MyoSuite).

**Target network** We investigate the performance benefits stemming from using a target network with the BRO agent. We present these results in Figure 16. Interestingly, we observe that the target network yields performance improvements only in specific environments.

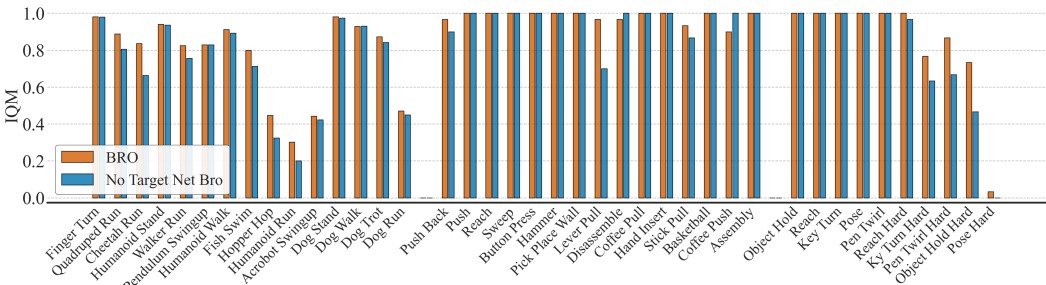

Figure 16: We compare BRO against BRO without target network. 10 seeds per task, $1M$ steps.

**More baselines** To evaluate BRO performance beyond maximum entropy objectives, we tested BRO and BRO (Fast) with a TD3 backbone. BRO with a SAC backbone slightly outperformed TD3, though TD3 remains a viable option. Furthermore, we compare BRO performance to three additional baselines. These results are presented in Figure 17.

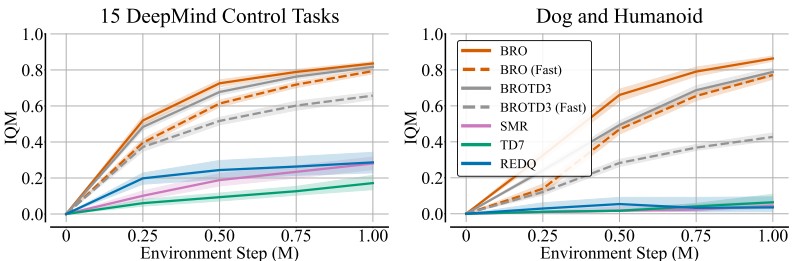

Figure 17: We run BRO and BRO (Fast) with a TD3 backbone (BROTD3). 10 seeds.

**Longer training** We expanded BRO training beyond 1M environment steps, although in a single-task setup. We trained BRO and BRO (Fast) for 3M and 5M steps respectively on 7 Dog and Humanoid tasks and compared them to TD-MPC2 and SR-SAC. BRO significantly outperforms these baselines and notably almost solves the Dog Run tasks at 5M steps (achieving over 80% of possible returns). We show the 3M results in Figure 18.

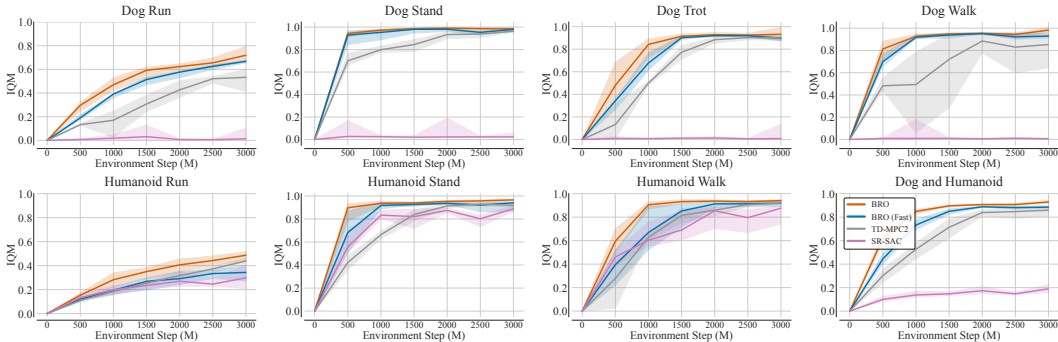

Figure 18: We run BRO on complex tasks for 3M steps. 5 seeds.

**Image-based tasks** To analyze the impact of BroNet on image-based RL tasks, we experiment with 3 tasks from the Atari 100k (Bellemare et al., 2013) benchmark. Here, we changed the regular

Q-network of the SR-SPR (RR=2) model (D'Oro et al., 2022) to a BroNet, and considered changing the reset schedules to better fit the plasticity of the BroNet model. As depicted in the Table below, applying BroNet to discrete, image-based tasks is a promising avenue for future research.

Table 8: We replace the Q-network in SR-SPR with BroNet, while keeping the standard convolutional encoder. We test two values of reset interval (RI) and shrink-and-perturb (SP) and find that these hyperparameters impact the performance of the BroNet agent. 5 seeds.

| Task | SR-SPR (RI=20k;SP=0.8) | SR-SPR-BroNet (RI=20k;SP=0.8) | SR-SPR-BroNet (RI=20k;SP=0.5) | SR-SPR-BroNet (RI=5k;SP=0.8) |
|---|---|---|---|---|
| Pong | -10.5 | 4.8 | 10.2 | -12.0 |
| Seaquest | 714.7 | 399.0 | 420.7 | 782.0 |
| Breakout | 24.5 | 13.5 | 13.7 | 33.3 |

**Multi-Task RL**  Finally, we evaluate whether the increased model size improves the agent's capability in a multi-task setup (Yu et al., 2020; Hansen et al., 2023). To this end, we compare the BRO (Fast) to SAC on the MT50 benchmark from MetaWorld, which comprises 50 different tasks. To accommodate the task diversity, we increase the width of all models twofold and pass a one-hot vector representing the task identifier in the network input. Otherwise, both algorithms are run with the same hyperparameters as the main experiments. We present the results in Figure 19.

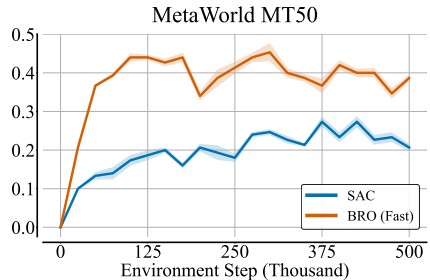

Figure 19: We compare BRO (Fast) with SAC on the multi-task benchmark. 3 seeds

# F  Training Curves

We present the aggregated performance of BRO compared to other baselines at Dog & Humanoid tasks, DeepMind Control Suite, Metaworld and MyoSuite in Figure 20 together with summarized performance results at $100k$, $200k$, $500k$ and $1M$ steps in Table 9. The performance on each of the 40 individual tasks in shown in Figures 21, 23, 22.

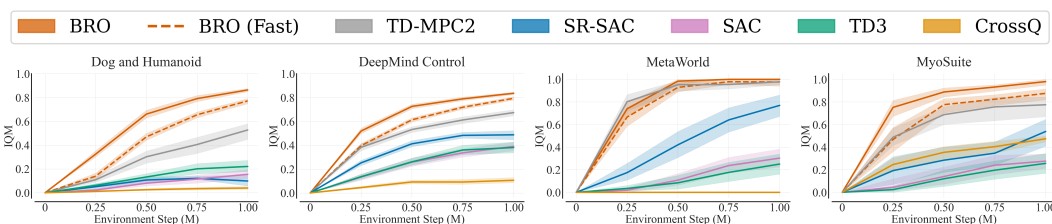

Figure 20: BRO **aggregated** performance over $1M$ steps on 40 **tasks** from DeepMind Control Suite, MetaWorld and MyoSuite. $Y$-axis represents the IQM of normalized rewards.

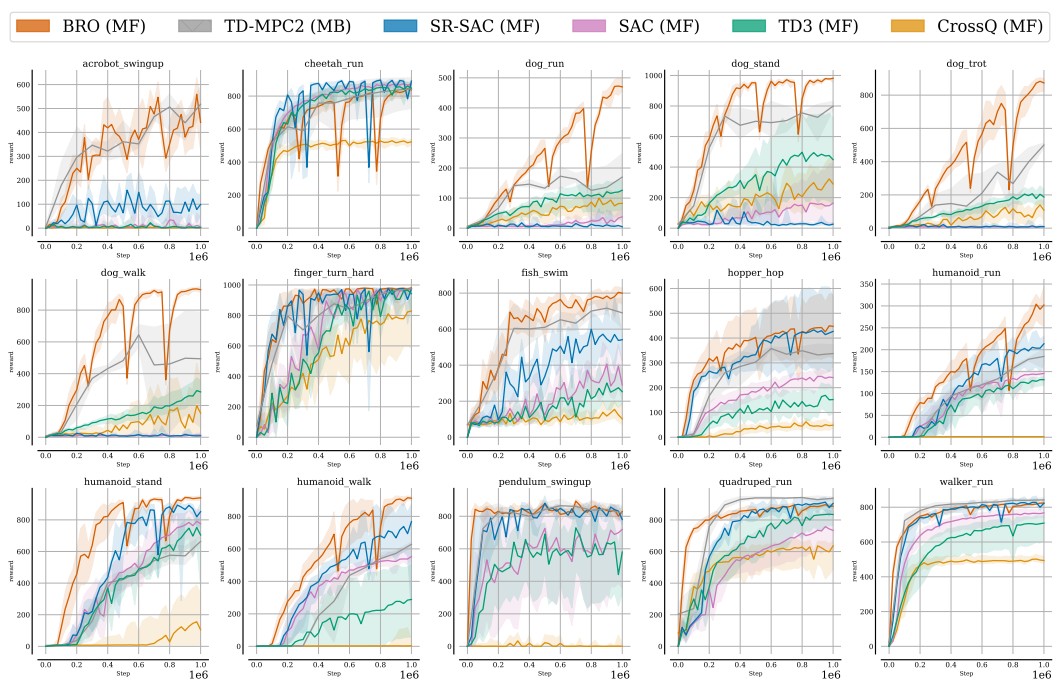

Figure 21: Results of 15 tasks from **DeepMind Control Suite** for BRO and other baselines run for $1M$ steps. We present the IQM of rewards and $95\%$ confidence intervals.

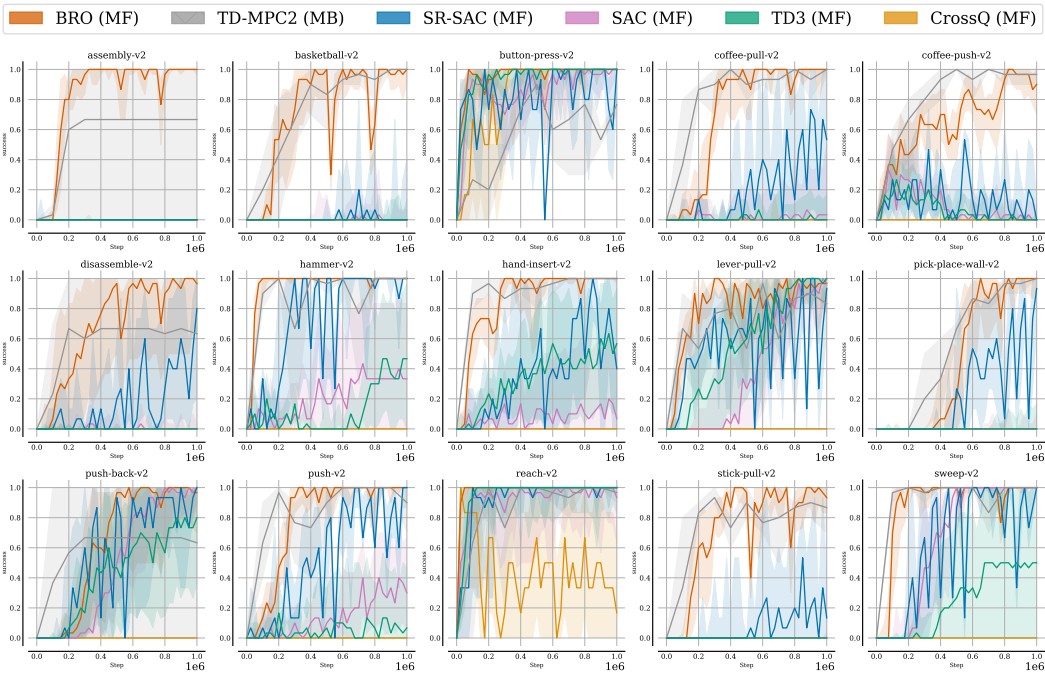

Figure 22: Results of 15 tasks from **MetaWorld** for BRO and other baselines run for $1M$ steps. We present the IQM of success rate and $95\%$ confidence intervals.

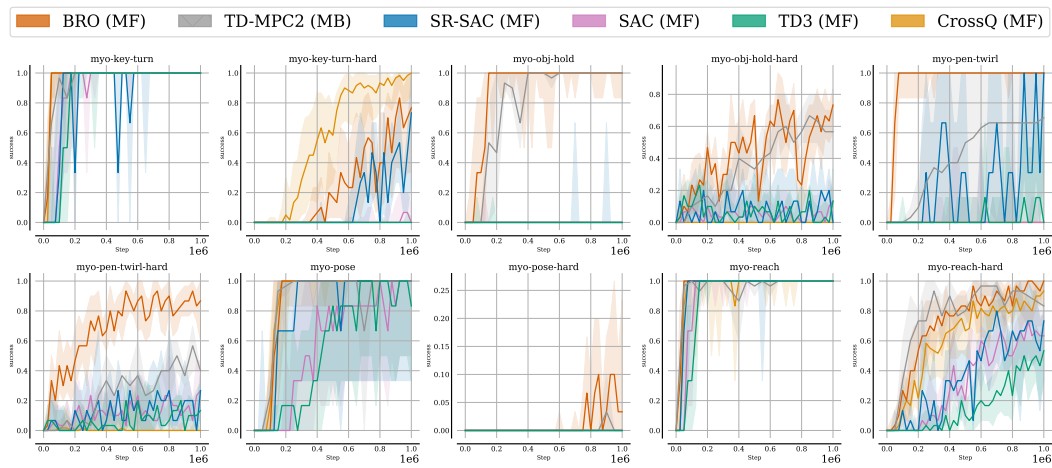

Figure 23: Results of 10 tasks from **MyoSuite** for BRO and other baselines run for $1M$ steps. We present the IQM of success rate and $95\%$ confidence intervals.

Table 9: Summary of IQM results of BRO and other agents evaluated on $40$ tasks from DeepMind Control Suite, Metaworld and MyoSuite achieved at $100k$, $200k$, $500k$ and $1M$ steps. BRO achieves **better results** than other state-of-the-art agents (both model-based and model-free) while featuring great sample efficiency.

| Step | BRO | BRO (Fast) | TD-MPC2 | SR-SAC | SAC | TD3 | CrossQ |
|---|---|---|---|---|---|---|---|
| AGGREGATED 40 TASKS | | | | | | | |
| $100k$ | **0.254** | 0.113 | 0.204 | 0.046 | 0.007 | 0.008 | 0.004 |
| $200k$ | **0.560** | 0.384 | 0.519 | 0.083 | 0.043 | 0.054 | 0.009 |
| $500k$ | **0.862** | 0.772 | 0.745 | 0.373 | 0.167 | 0.157 | 0.037 |
| $1M$ | **0.945** | 0.878 | 0.842 | 0.595 | 0.316 | 0.294 | 0.042 |
| DEEPMIND CONTROL SUITE | | | | | | | |
| $100k$ | **0.230** | 0.123 | 0.128 | 0.089 | 0.030 | 0.046 | 0.031 |
| $200k$ | **0.442** | 0.282 | 0.332 | 0.195 | 0.088 | 0.091 | 0.038 |
| $500k$ | **0.726** | 0.613 | 0.532 | 0.412 | 0.259 | 0.261 | 0.092 |
| $1M$ | **0.836** | 0.794 | 0.673 | 0.487 | 0.391 | 0.381 | 0.106 |
| METAWORLD | | | | | | | |
| $100k$ | 0.247 | 0.113 | **0.452** | 0.046 | 0.000 | 0.000 | 0.000 |
| $200k$ | 0.642 | 0.571 | **0.835** | 0.062 | 0.018 | 0.047 | 0.000 |
| $500k$ | **0.984** | 0.929 | 0.952 | 0.421 | 0.108 | 0.084 | 0.000 |
| $1M$ | **1.000** | 0.976 | 0.978 | 0.769 | 0.303 | 0.250 | 0.000 |
| MYOSUITE | | | | | | | |
| $100k$ | **0.392** | 0.124 | 0.088 | 0.000 | 0.000 | 0.000 | 0.020 |
| $200k$ | **0.748** | 0.400 | 0.394 | 0.015 | 0.024 | 0.008 | 0.140 |
| $500k$ | **0.888** | 0.776 | 0.688 | 0.285 | 0.140 | 0.120 | 0.354 |
| $1M$ | **0.980** | 0.876 | 0.775 | 0.538 | 0.276 | 0.256 | 0.474 |
| DOG & HUMANOID | | | | | | | |
| $100k$ | **0.038** | 0.008 | 0.014 | 0.007 | 0.006 | 0.013 | 0.011 |
| $200k$ | **0.238** | 0.056 | 0.058 | 0.024 | 0.015 | 0.040 | 0.013 |
| $500k$ | **0.661** | 0.469 | 0.302 | 0.107 | 0.080 | 0.133 | 0.025 |
| $1M$ | **0.864** | 0.772 | 0.527 | 0.099 | 0.155 | 0.221 | 0.040 |

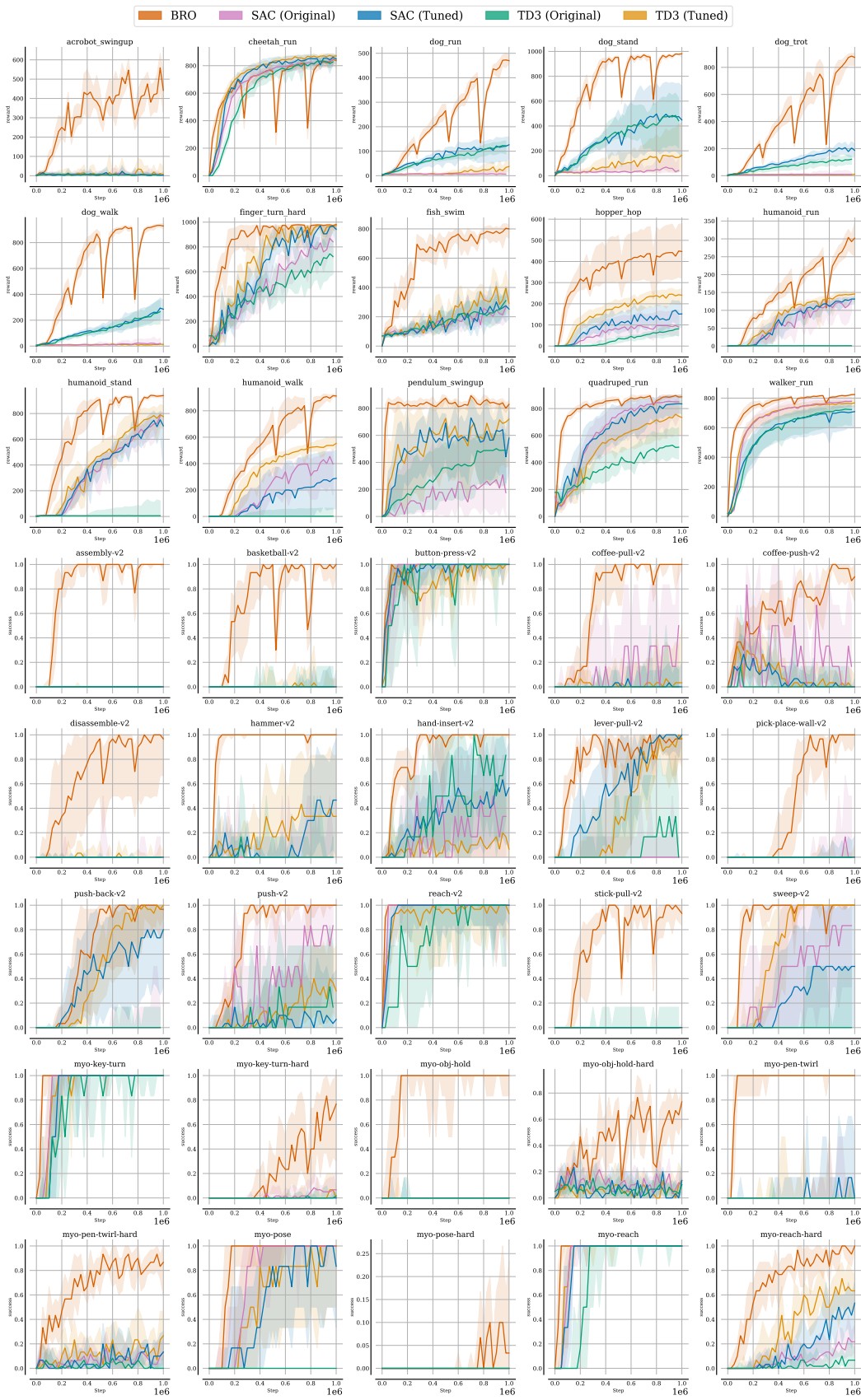

Figure 24: Results of different variants of SAC and TD3 on 40 tasks.

