# OpenReview forum: "Bigger, Regularized, Optimistic: scaling for compute and sample efficient continuous control"
_NeurIPS.cc/2024/Conference — NeurIPS 2024 spotlight_

### Official Review · Reviewer_TNcA · 2024-06-26

**Soundness:** 3
**Presentation:** 2
**Contribution:** 3
**Rating:** 7
**Confidence:** 4

**Summary:**

This paper investigates the sample efficiency problem in continuous control. The authors propose the BRO algorithm, i.e., Bigger, Regularized, Optimistic. The authors find that strong regularization allows for effective scaling of the critic networks, which, paired with optimistic exploration, leads to quite good performance. BRO achieves strong performance on numerous continuous control benchmarks, and is the first model-free reinforcement learning algorithm that can learn meaningful performance in DMC dog and humanoid tasks.

**Strengths:**

- this paper is easy to follow and easy to understand

- the studied topic is important to the RL community. It is vital to develop stronger and more powerful model-free RL algorithms for continuous control problems

- despite that this work combines numerous previous well-developed tricks and strategies, the authors selectively incorporate them into one framework and demonstrate that such design choice incurs quite good performance.

- the experiments are extensive and solid

- the developed BRO algorithm is the first model-free RL algorithm that can achieve meaningful performance in DMC dog and humanoid tasks

**Weaknesses:**

this paper has the following drawbacks,

- the quality of the figures could be significantly improved. Please try to export pdf with matplotlib instead of taking screenshots by convention.

- BRONet seems to be a simplified version of ResNet. I do not seem to observe any significant network architecture difference between them. The authors should not over-claim on the network architecture. Any clarifications here?

- It is often unclear how the figures are plotted and which environments they cover, e.g., Figure 4, Figure 6. This should not be vague and ought to be clearly stated in the main text

- I am a bit concerned with the claim that *Algorithmic improvements matter less as the scale increases* (Line 184). Do you think that this is always correct? One should focus more on scaling instead of algorithmic improvements in the context of RL?

- missing baselines and references. The authors should compare against some other recent strong model-free RL algorithm, e.g., TD7 [1]. Meanwhile, the authors should cite the REDQ [2] paper when referring to replay ratios. REDQ should be included as a baseline approach in the paper. Moreover, a recent paper introduces sample multiple reuse (SMR) [3] that updates a fixed batch multiple times to boost sample efficiency. I think this can be a very relevant work and should be included and discussed in the paper. Also, the authors write that they introduce another actor network, and incorporate regularization techniques into critics, it reminds me of another work, DARC [4], where they leverage double actors for exploration and introduce critic regularization for better performance. It should be included in the paper.

[1] For sale: State-action representation learning for deep reinforcement learning

[2] Randomized Ensembled Double Q-learning: Learning Fast Without a Model

[3] Off-Policy RL Algorithms Can be Sample-Efficient for Continuous Control via Sample Multiple Reuse

[4] Efficient Continuous Control with Double Actors and Regularized Critics

Despite the aforementioned drawbacks, this is a solid paper that may have a potentially large impact on the RL community. I would be happy to reconsider the score if the aforementioned flaws are addressed during the rebuttal phase.

**Questions:**

- how do you expect BRO to be applied in the discrete control tasks? Will BRO beat other stronger MCTS-based approaches in Atari games? Any comments here?

- can you elaborate more on explaining why scaling the actor network is not effective?

**Limitations:**

The authors have a good discussion of the potential limitations of this work in the main text.

---

> ### Author Rebuttal · Authors · 2024-08-06
>
> We thank the reviewer for their time reviewing our work and the suggestions on how to improve it. We are also very pleased that the reviewer found the experimental section solid. We leave our rebuttal below:
>
> >missing baselines and references...
>
> We thank the reviewer for suggesting these algorithms. In response, we added an experiment comparing BRO and BRO (Fast) to TD7, SMR, and RedQ on the 15 DMC tasks from our original evaluations. Given the limited time and the marginal performance improvements reported over SAC/TD3, we excluded DARC but cited and discussed it in the related work section. Our findings show that BRO and BRO (Fast) significantly outperform the additional baselines, and we detail these results in the rebuttal PDF we uploaded.
>
> >The quality of the figures could be significantly improved...
>
> We thank the reviewer for noticing that - indeed some of our figures were embedded as PNG files. We changed all figures to vector PDFs.
>
> >BRONet seems to be a simplified version of ResNet…
>
> BroNet uses layer normalization and adds an extra layer normalization before the first residual stream, as shown in Figure 12. This addition boosts performance by over 20% in DMC tasks, aligning with prior studies like [1] and [2] that reported declines when applying standard ResNets to Atari and DMC benchmarks. We named this configuration "BroNet" to highlight these crucial design choices. Additional offline RL experiments demonstrating BroNet's effectiveness are detailed in our joint response and Figure 3 of the rebuttal PDF. At the same time, we agree that it is crucial not to present BroNet as a new neural architecture but as an effective integration of existing modules for RL applications. We believe that naming this configuration aids in distinguishing specific architectures within the community. We added this discussion to the Method section. We hope this meets the reviewer’s expectations, though we are open to revising the presentation in the manuscript as needed.
>
> >It is often unclear how the figures are plotted and which environments they cover, e.g., Figure 4, Figure 6...
>
> We thank the reviewer for noticing that. We checked all figures for links to task lists and added them when missing. We use two task sets in our experiments: all 40 tasks from DMC, MW, and MYO, and 10 tasks from DMC and MW. The majority of experiments (e.g. Figure 6) were performed on all 40 tasks.
>
> >I am a bit concerned with the claim that Algorithmic improvements matter less...
>
> We thank the reviewer for this feedback. We acknowledge that scaling may not always be beneficial, such as in tasks with sparse rewards and complex exploration. However, our main takeaway is that scaling, alongside optimization techniques like layer normalization and replay ratio, can significantly enhance performance, often surpassing RL-specific algorithmic advancements (e.g., Figures 4 & 7). Many recent algorithms focus solely on these specific improvements while maintaining traditional architectures with two layers of 128-256 units (e.g., RedQ, SMR, TD7, SR-SAC [4]). We believe the synergy between algorithmic and scaling enhancements warrants more attention in future research. While we agree that algorithmic improvements are crucial for the field's progress, we aimed to suggest they matter "less" rather than "not at all." We are open to modifying the wording based on the reviewer’s feedback.
>
> >how do you expect BRO to be applied in the discrete control tasks...?
>
> We focused exclusively on continuous control because there are significant differences in best practices between continuous and discrete algorithms. For example, continuous algorithms usually require more mechanisms to curb overestimation [3] or different replay ratios [4]. However, we are happy to announce additional experiments in the Atari benchmark, where we use 3 tasks and augment the SR-SPR model with BroNet and find BroNet promising in these tasks. We summarize these results in the joint response. Due to the large computational demands associated with MCTS-style algorithms, we leave comparing BRO to those for future work. We summarize these results in the joint response and the rebuttal PDF.
>
> >can you elaborate more on explaining why scaling the actor network is not effective?
>
> We interpret this result to be consistent with the previous works showing that off-policy actor-critic algorithms are more “critic centric” (e.g. [3] showing that actor can be updated two times less frequently, or [4] showing that resetting critic is more important than resetting actor). We hypothesize that the critic learning is thus more complex because it models the dimension of actions as well (and policy changes only wrt. state). Furthermore, since the policy optimizes for critic output, the policy complexity is inherently capped by the expressiveness of the critic.
>
> We thank the reviewer again for their time, insights, and suggestions. Based on these we made several additions to our manuscript, which we describe in detail in the joint response. In particular, we hope that the additional baselines and added references address the issue of “missing baselines and references”. We also hope that the added results in the offline RL and discrete image-based RL benchmarks increase the reviewer’s confidence in the BroNet utility. If so, we kindly ask the reviewer to consider adjusting the initial score of our work.
>
> [1] Schwarzer, Max, et al. "Bigger, better, faster: Human-level atari with human-level efficiency."
>
> [2] Bjorck, Nils, et al. "Towards deeper deep reinforcement learning with spectral normalization."
>
> [3] Fujimoto, S., et al. Addressing function approximation error in actor-critic methods.
>
> [4] D'Oro, Pierluca, et al. "Sample-efficient reinforcement learning by breaking the replay ratio barrier."

---

> > ### Comment · Reviewer_TNcA · 2024-08-08
> >
> > I thank the authors for the detailed rebuttal experiments. Please find the comments below,
> >
> > > missing baselines and references
> >
> > It is good to see experiments on TD7 and SMR. The experiments can be added to the paper, maybe in the appendix. Please also consider citing the references which I believe can further strengthen this work.
> >
> > > figure quality
> >
> > Yes, please do change all figures to vector PDFs
> >
> > > BRONet and ResNet
> >
> > Thanks for the clarification. I would recommend the authors not to over-claim the network architecture. Please consider revising the presentation to properly position the contribution of your work
> >
> > > how the figures are plotted and which environments they cover
> >
> > Please make them clearer in the final version. As commented, these should not be vague.
> >
> > > the claim that Algorithmic improvements matter less as the scale increases
> >
> > Thanks for the clarification. I think the authors should incorporate the rebuttal clarifications into the revision and have a more careful discussion on this.
> >
> > > BRO in Atari games
> >
> > Thank you for the additional experiments. I understand that the MCTS-based method can be computationally inefficient. I would recommend the authors take a look at the EfficientZero [1] algorithm, which is quite efficient and runs very fast on Atari 100k games. The performance of EfficientZero is also quite good. However, I understand that discrete control is not the focus of this work, and it is okay to see that BRONet beats SR-SPR on some Atari games.
> >
> > [1] Mastering atari games with limited data
> >
> > > why scaling the actor network is not effective
> >
> > Thanks for the clarification.
> >
> > All in all, this is a solid work. It is my hope that the authors can revise the manuscript based on my comments and suggestions. Considering its potential impact on the RL community and the rebuttal addresses all my concerns, I feel more confident to vote for accepting this paper. I am happy to increase my score from 5 to 7. Congratulations!

---

> > > ### Author Response · Authors · 2024-08-08
> > > **We thank the reviewer for their prompt response**
> > >
> > > We thank the reviewer for their prompt response and for increasing the score of our manuscript. We commit to the careful implementation of the above in our final version of the paper.
> > >
> > > We are also happy to answer new questions if any arise.
> > >
> > > Best regards,
> > > Authors

---

### Official Review · Reviewer_7YXC · 2024-07-11

**Soundness:** 3
**Presentation:** 3
**Contribution:** 4
**Rating:** 7
**Confidence:** 4

**Summary:**

This paper investigates how reinforcement learning (RL) can benefit from parameter scaling. The authors introduce BroNet, a variant of ResNet with LayerNorm, as a well-regularized network structure for the critic that improves performance when scaled. They also find that when the critic is properly regularized, the common Clipped Double Q-learning trick can be replaced with optimistic exploration, further boosting sample efficiency. These findings are combined into a new algorithm called BRO.

To demonstrate the efficiency of their approach, the authors conduct extensive experiments on 40 challenging tasks across 3 benchmark suites. They also provide comprehensive ablation studies to justify their design choices.

**Strengths:**

- The paper addresses the important and timely topic of scaling in model-free off-policy RL.
- It offers extensive large-scale studies on various design choices, providing valuable insights to the RL community.
- The proposed BRO algorithm shows promising results across a wide range of tasks.

**Weaknesses:**

As acknowledged by the authors, the study primarily focuses on state-based off-policy RL. The transferability of their conclusions to other domains such as image-based problems and offline RL remains unclear, which limits the paper's broader impact.

**Questions:**

- The benefits of scaling the critic appear to saturate after 5M parameters. What are your projections for further scaling? Do you anticipate additional benefits beyond this point, or do you believe there are fundamental limitations?
- Could you provide more details on the evaluation metrics used, particularly for the MetaWorld benchmark? Given the various reporting methods in the literature for MetaWorld success rates, it would be helpful to clarify which specific metric was employed in this study.

**Limitations:**

Yes, the authors have discussed the limitation in section 5.

---

> ### Author Rebuttal · Authors · 2024-08-06
>
> We thank the reviewer for their time and valuable suggestions. We are also happy that the reviewer found the insights provided by our manuscript valuable. Please share our rebuttal below:
>
> >As acknowledged by the authors, the study primarily focuses on state-based off-policy RL. The transferability of their conclusions to other domains such as image-based problems and offline RL remains unclear, which limits the paper's broader impact.
>
> We thank the reviewer for pointing us towards these domains. As described in the joint response, we are happy to announce two major additions to our manuscript.
>
> Firstly, we add experiments on two offline RL benchmarks: AntMaze (6 tasks); and Adroit (9 tasks) [1]. We design these experiments to test the effectiveness of the naive application of BroNet to popular offline approaches. To this end, we run Behavioral Cloning (BC) in pure offline, Implicit Q-Learning (IQL) offline + fine-tuning, and SAC with RLPD buffer online with offline data setup. We run all these algorithms with the default network and BroNet backbones. We find that the naive application of BroNet leads to performance improvements across all tested algorithms.
>
> Secondly, we add experiments in discrete image-based RL benchmark Atari 100k [2]. Similarly to offline, we evaluate the application of BroNet to the popular discrete RL algorithm SR-SPR. In this experiment, we substitute the regular critic network with a BroNet (without changing the convolutional encoder on other parts of the model like learning rate). We run 5 seeds on 3 tasks and find that BroNet can improve the performance of SR-SPR, but its effectiveness is dependent on the reset configuration and thus demands future work.
>
> We show these experimental results in the uploaded rebuttal PDF. Additionally, we added a passage to our limitations section where we state that the usefulness of BRO for offline/image-based setups should be studied further. We hope that these added experimental results are a valuable addition for readers interested in vision-based and offline control, as well as increase the reviewer’s confidence in the contribution of our manuscript.
>
> >The benefits of scaling the critic appear to saturate after 5M parameters. What are your projections for further scaling? Do you anticipate additional benefits beyond this point, or do you believe there are fundamental limitations?
>
> We believe that performance saturation beyond 5M parameters is due to the low complexity of the environments studied. Therefore, we anticipate that larger models will be beneficial for more challenging tasks, such as real-world robotics, multi-task learning, and image-based control in varied environments. Scaling is expected to become an increasingly significant tool in reinforcement learning. However, it is not a universal solution, and complex challenges like exploration may need approaches beyond just scaled SAC or BRO. We added this remark to Section 3.
>
> >Could you provide more details on the evaluation metrics used, particularly for the MetaWorld benchmark? Given the various reporting methods in the literature for MetaWorld success rates, it would be helpful to clarify which specific metric was employed in this study.
>
> We thank the reviewer for this suggestion. We added the following text to our manuscript detailing our evaluation method: “In the MetaWorld environment we follow the TD-MPC2 evaluation protocol. As such, the environment issues a truncate signal after 200 environment steps, after which we assess if the agent achieved goal success within the 200th step. We do not implement any changes to how goals are defined in the original MetaWorld and we use V2 environments”. We hope that this clarifies our evaluation procedure for MetaWorld tasks. Please inform us if otherwise.
>
> We thank the reviewer again for their valuable insights and suggestions on how to improve our manuscript. We think that implementing these suggestions resulted in additional value for readers, especially those interested in applying BRO to other domains than continuous control (e.g. offline, image-based, or discrete RL).
>
> [1] Fu, Justin, et al. "D4rl: Datasets for deep data-driven reinforcement learning."
>
> [2] Kaiser, Lukasz, et al. "Model-based reinforcement learning for atari."

---

> > ### Comment · Reviewer_7YXC · 2024-08-10
> > **Thanks for the additional results**
> >
> > I appreciate the author providing additional results on offline RL and image-based RL. However, the experiments seem to deviate from the goal of this paper. Changing the network from MLP to BroNet (with the same parameter size, I presume?) only shows the efficiency of BroNet other than "scaling also helps offline RL and image-based RL". The correct experiments should be also changing the size of BroNet like what you did in the main paper. Especially as you argue the the results saturate with 5M since the tasks are too simple, maybe you can prove that on the image-based tasks.
> >
> > I understand these experiments could be too expensive during the rebuttal phase. Thus, I would encourage you to provide them in the final version.

---

> ### Author Response · Authors · 2024-08-11
> **Thank you for the quick response and apologies for the confusion**
>
> We thank the reviewer for their quick response. We also apologize for not describing the additional experiments thoroughly enough and for the confusion that resulted.
>
> In the offline and Atari experiments we are using a scaled BroNet. Specifically, in the offline experiments, we substitute the standard MLP (2 layers of 256 units) with the standard BroNet variant that we use in BRO and BRO (fast) (i.e. 6 layers with 512 units). In the Atari experiments, we leave the convolutional encoder untouched and substitute the Q-network head (1 layer of 512 units) with a slightly smaller BroNet (i.e. 4 layers with 512 units). We used 4-layer BroNet because the Q-network head used in the original implementation has only a single layer. We hope this answer clarifies that the newly added experiments are in line with the results presented in the main body (i.e., that we present results of a BroNet with scaled parameter count).
>
> We also thank the reviewer for suggesting using >5M in the image-based setup. We will provide these results in the camera-ready version.
>
> Regards,
> Authors

---

### Official Review · Reviewer_zd6E · 2024-07-17

**Soundness:** 3
**Presentation:** 4
**Contribution:** 4
**Rating:** 8
**Confidence:** 4

**Summary:**

The paper studies how to scale up RL algorithms in the continuous domain and introduces the BRO (Bigger, Regularized, Optimistic) algorithm, designed to enhance sample efficiency with (relatively) large models. The authors conduct extensive experiments to verify the effectiveness of factors like replay ratio, regularizations, optimistic exploration, and quantile Q-values when scaling up RL algorithms. The findings from these extensive experiments lead to the novel BRO algorithm, which consists of a novel architecture with proper regularization and exploration. Empirical results demonstrate that BRO achieves state-of-the-art performance on 40 complex tasks from the DeepMind Control, MetaWorld, and MyoSuite benchmarks, outperforming leading model-based and model-free algorithms, especially in the challenging Dog and Humanoid tasks.

**Strengths:**

1. This paper tackles an important problem of scaling up in reinforcement learning, especially in continuous action domains.

2. The authors conduct extensive experiments on the effects of different methods on scaling up, which I found very informative.

3. The proposed algorithm, BRO, achieves strong empirical performances on various domains, especially on the Dog & Humanoid domains.

4. The paper is well-structured and well-written.

**Weaknesses:**

Usually, scaling up benefits more when a large amount of data is available, where large models can lead to positive transfer or generalization across different tasks. However, the current setup is the same as the standard setting, where the agent is trained for 1M steps on each task separately. The work would be more significant if the model is trained on and can be transfered across different tasks.

**Questions:**

See the weakness section. Is there any evidence that the proposed method can benefit from training on diverse tasks?

**Limitations:**

The authors discussed their limitations well in the paper.

---

> ### Author Rebuttal · Authors · 2024-08-06
>
> We thank the reviewer for their time and valuable feedback regarding our work. We are very pleased that the reviewer found our results on scaling interesting. Please find our rebuttal below.
>
> >Usually, scaling up benefits more when a large amount of data is available, where large models can lead to positive transfer or generalization across different tasks. However, the current setup is the same as the standard setting, where the agent is trained for 1M steps on each task separately. The work would be more significant if the model is trained on and can be transferred across different tasks.
>
> We thank the reviewer for their excellent question related to generalization across different tasks. In this work, we intentionally focused on the single-task setup because it is the standard approach in previous work proposing base agents [1, 2]. Furthermore, it provides a clear and isolated setting to study the properties of RL algorithms and serves as a necessary foundational step before tackling multi-task learning. During the limited rebuttal period, we enhanced our single-task analysis by the experiments described in the single-task setup (e.g. longer training, offline, image-based, and BRO with TD3 backbone), further confirming that BRO is an attractive option for future studies. Nevertheless, conducting informative multi-task or continual learning experiments requires significantly more work, as highlighted by e.g. [3].
>
> However, we commit to running simple preliminary multi-task experiments for the camera-ready version. We also added this discussion to the future work section.
>
> >Is there any evidence that the proposed method can benefit from training on diverse tasks?
>
> There are generic arguments that suggest that BRO could perform well in a multi-task learning environment. For example, the increased network capacity combined with robust regularization techniques should theoretically decrease catastrophic forgetting. Moreover, these design choices should also help in reducing overfitting to individual tasks, thus increasing generalization across tasks. We do not have any insights into the quality of representations learned by BRO in a multi-task setup, but such research could be interesting when tackling multi-task RL.
>
> We thank the reviewer again for their time and insights. We are happy to answer further questions if any arise. We also welcome further discussion on the inclusion of preliminary multi-task results in the camera-ready version of our paper.
>
> [1] Hessel, Matteo, et al. "Rainbow: Combining improvements in deep reinforcement learning."
>
> [2] Schwarzer, Max, et al. "Bigger, better, faster: Human-level atari with human-level efficiency."
>
> [3] Wolczyk, Maciej, et al. "Disentangling transfer in continual reinforcement learning."

---

> > ### Comment · Reviewer_zd6E · 2024-08-14
> >
> > Thank you for the detailed response. I believe the additional experiment strengthens the paper.

---

### Author Rebuttal · Authors · 2024-08-06

We thank the reviewers for their insightful feedback. We are pleased that our work was well received, and that all reviewers recognized the potential significance of our work for the RL community, the scope of our experiments, and the significant performance improvements our method provides over previous approaches. Following the reviewers' suggestions, we have made additions to our manuscript that we believe further increase its value. We have included graphs of the new results in the rebuttal PDF and summarize them here:

**New baselines** - Following the recommendation of reviewer TNcA, we have added three new baselines: TD7 (original codebase), RedQ (JaxRL), and SMR (JaxRL) and tested them on 15 DMC tasks. In these experiments, our proposed BRO achieves 250-350% better performance than the suggested algorithms. These results are displayed in Figure 1 of the rebuttal PDF.

**Extended training** - Following reviewer zd6E's suggestion, we expanded BRO training beyond 1M environment steps, although in a single-task setup. We trained BRO and BRO (Fast) for 3M and 5M steps respectively on 7 Dog and Humanoid tasks and compared them to TD-MPC2 and SR-SAC. BRO significantly outperforms these baselines and notably almost solves the Dog Run tasks at 5M steps (achieving over 80% of possible returns). We show the 3M results in Figure 2 of the rebuttal PDF.

**Offline RL benchmark** - As suggested by reviewer 7YXC, we have added experiments on two offline RL benchmarks: AntMaze (6 tasks) and Adroit (9 tasks) [1]. We tested three scenarios: pure offline (comparing vanilla Behavioral Cloning (BC) to BroNet-based BC), offline with fine-tuning (comparing vanilla IQL [2] to BroNet-based IQL), and online with offline data (comparing vanilla SAC to BroNet-based SAC). Using BroNet led to noticeable improvements for all learners, as depicted in Figure 3 of the rebuttal PDF.

**Image-based benchmark** - Following Reviewer 7YXC, we added experiments on 3 tasks from the Atari 100k [3] benchmark. Here, we changed the regular Q-network of the SR-SPR (RR=2) model [4] to a BroNet, and considered changing the reset schedules to better fit the plasticity of the BroNet model. As depicted in Table 1 of the uploaded PDF, applying BroNet to discrete, image-based tasks is a promising avenue for future research.

**BRO + TD3** - To evaluate BRO performance beyond maximum entropy objectives, we tested BRO and BRO (Fast) with a TD3 backbone across 15 DMC tasks. BRO with a SAC backbone slightly outperformed TD3, though TD3 remains a viable option. This result might be helpful for practitioners interested in applying BRO to models with TD3 backbone, such as the image-based SOTA algorithm DrM [5]. These findings are illustrated in Figure 1 of the rebuttal PDF.

We hope that these results show that our approach to scaling seems to be promising in other branches of RL as well, and will ultimately prove to be helpful for readers interested in problems beyond continuous control. We believe that these substantially increased the quality of our manuscript, and again, we are grateful to the reviewers for their suggestions. We invite the reviewers to inspect the new results in the uploaded rebuttal PDF and are happy to answer any further questions.


[1] Fu, Justin, et al. "D4rl: Datasets for deep data-driven reinforcement learning."

[2] Kostrikov, Ilya, Ashvin Nair, and Sergey Levine. "Offline reinforcement learning with implicit q-learning."

[3] Kaiser, Lukasz, et al. "Model-based reinforcement learning for atari."

[4] Schwarzer, Max, et al. "Data-efficient reinforcement learning with self-predictive representations."

[5] Xu, Guowei, et al. "Drm: Mastering visual reinforcement learning through dormant ratio minimization."

---

> ### Author Response · Authors · 2024-08-14
> **We thank for the rebuttal**
>
> We sincerely thank the reviewers for their valuable feedback and insightful questions during the discussion phase. We are confident that incorporating their suggestions has significantly enhanced the quality and value of our manuscript.
>
> We will be ready to answer new questions if any arise.
>
> Best regards,
> Authors

---

### Decision · Program_Chairs · 2024-09-25

**Decision:**

Accept (spotlight)

**Comment:**

The paper makes improves SOTA for deep Reinforcement Learning RL (both model-free and model-based).  This is achieved using well-known techniques (using big critic networks, regularisation (layer normalisation, weight decay) and optimism in the face of uncertainty. Overall, this is a well-executed empirical paper. The most important component of this type of paper (empirical evaluation methodology) seems solid. Crucially, the submission provides a link to code used to generate the results, which is crucial for this type of paper.